# A large invasive consumer reduces coastal ecosystem resilience by disabling positive species interactions

Marc J. S. Hensel[1,2✉], Brian R. Silliman[2], Johan van de Koppel[3,4], Enie Hensel[5], Sean J. Sharp[6], Sinead M. Crotty[7] & Jarrett E. K. Byrnes[1]

Invasive consumers can cause extensive ecological damage to native communities but effects on ecosystem resilience are less understood. Here, we use drone surveys, manipulative experiments, and mathematical models to show how feral hogs reduce resilience in south-eastern US salt marshes by dismantling an essential marsh cordgrass-ribbed mussel mutualism. Mussels usually double plant growth and enhance marsh resilience to extreme drought but, when hogs invade, switch from being essential for plant survival to a liability; hogs selectively forage in mussel-rich areas leading to a 50% reduction in plant biomass and slower post-drought recovery rate. Hogs increase habitat fragmentation across landscapes by maintaining large, disturbed areas through trampling of cordgrass during targeted mussel consumption. Experiments and climate-disturbance recovery models show trampling alone slows marsh recovery by 3x while focused mussel predation creates marshes that may never recover from large-scale disturbances without hog eradication. Our work highlights that an invasive consumer can reshape ecosystems not just via competition and predation, but by disrupting key, positive species interactions that underlie resilience to climatic disturbances.

---

[1] Department of Biology, University of Massachusetts Boston, Boston, MA, USA. [2] Nicholas School for the Environment, Duke University, Durham, NC, USA. [3] NIOZ Royal Netherlands Institute for Sea Research, Department of Estuarine and Delta Systems, Utrecht University, Utrecht, Netherlands. [4] Groningen Institute for Evolutionary Life Sciences (GELIFES), University of Groningen, Groningen, Netherlands. [5] Department of Applied Ecology, North Carolina State University, Raleigh, NC, USA. [6] School for Environment and Sustainability, University of Michigan, Ann Arbor, MI, USA. [7] Department of Environmental Engineering, University of Florida, Gainesville, FL, USA. ✉email: Marc.Hensel001@umb.edu

Non-native invasive species have spread to ecosystems around the world due to human activities, causing extinctions, biodiversity declines, and habitat loss[1–5]. Some of the strongest examples of these effects come from systems where invasions ripple through sets of keystone species interactions[6–10]. When invasive consumers integrate into native food webs, such as in plant–pollinator interactions[11–14], introduced beavers in South America[15], or in the Great Lakes where invasive predatory zooplankton has caused over $300 million USD in ecosystem service loss from water clarity and phosphorus loading[16], changes to native species interactions, directly and indirectly, degrade biodiversity, habitat structure, and ecosystem services[3,15,17–19]. While invasive species effects on trophic interactions are well-documented, management plans must begin to take a wider approach to factor in how invasive consumers alter non-trophic interactions that regulate ecosystem structure and resilience such as habitat cascades and non-feeding mutualisms[20].

Many terrestrial and aquatic ecosystems are strongly dependent on non-trophic positive interactions for recovery and resilience because facilitating species buffer abiotic stress, widen niches, increase population densities, and enhance ecosystem multifunctionality[20–25]. For example, a tree-epiphyte mutualism greatly increases whole arboreal arthropod community density and diversity in oak trees[26], and a seagrass–mussel–bacteria mutualism allows the formation of extensive seagrass ecosystems in the tropics[27]. Likewise, native beavers facilitate whole-pond communities by building dams, thereby increasing the diversity of microbial, plant, insect, bird, and fish communities and driving multiple ecosystem functions[28,29]. Positive interactions like these are predicted to become more important as stress increases within ecosystems due to global change[20], a result supported by a recent meta-analysis showing plant communities shift toward positive interactions when stress increases[30]. Thus, it is imperative to understand the disruptive forces that could unravel crucial positive interactions that underlie ecosystems' ability to recover from disturbance.

Coastal ecosystems in particular face increasing pressure from global change effects like drought, high storm severity, and frequency, and sea-level rise that often combine with local-scale habitat destruction, invasions, and eutrophication[31–37]. The coasts have remained among the most productive and economically valuable habitats[38,39] because long-term coastal ecosystem resilience is powered by positive interactions like facilitation cascades and mutualisms that increase biodiversity, speed recovery, and drive multifunctionality across landscapes[24,40–43]. As novel physical and biotic factors change coastal ecosystems and coastal human populations continue to rise, the conditions that threaten coastal protection from rising oceans are now relevant to millions of people[44].

In salt marshes of the southeastern U.S., non-trophic, mutualistic interactions are necessary to resist and recover from intense disturbance events. Over the past two decades, large-scale disturbances such as extreme drought, runaway consumer grazing, storms, and wrack deposition have caused expansive losses of the foundation species *Spartina alterniflora* (hereafter cordgrass) that forms monoculture seascapes that structurally define intertidal marsh habitat[45–47]. The impact of these disturbances has been mitigated by a keystone mutualism between cordgrass and the ribbed mussel, *Geukensia demissa* which only covers 0.1–12% of the total marsh surface, primarily in aggregated mounds[24,48]. In this mutualism, cordgrass facilitates mussels by providing settlement substrate and reducing temperatures via shading, while mussels in mounds enhance grass success, attract bioturbating burrowing crabs (e.g., *Uca sp.*), and protect grass from drought through amelioration of edaphic soil stress[24,40,48,49]. When drought-driven die-off of southeastern U.S. marshes occurs in an

area (from 1 to 100 s of km$^2$), cordgrass living on mussel mounds has a 98% chance at survival, while those not living on mussel mounds have a 0.01% chance[40]. Once disturbance subsides, this remnant cordgrass–mussel patches become nuclei for marsh recovery during non-drought or wet years[40,49]. With mussels present, large die-off areas recover in 2–10 years as opposed to 80 years when mussels are not present[40]. Thus, despite the growing threat of die-off, southeastern U.S. salt marshes remain one of the most productive ecosystems per unit area in the world[50] because these positive species interactions provide remarkable resilience[40,41,51].

In these southeastern U.S. marshes, the presence of a large invasive omnivore, the feral hog *Sus scrofa*, could threaten the resilience of the cordgrass–mussel mutualism. Hogs are well-known for the uprooting vegetation and consumption of small, ground-dwelling animals (e.g., insects and worms) that drastically change plant and animal communities in many terrestrial and aquatic habitats[52–54]. A survey of hog feces in marshes of the Sapelo Island National Estuarine Research Reserve found that 80% of hog feces contains mussel shells (Supplementary Table 1), suggesting that hogs frequently come into marshes primarily not necessarily to consume vegetation but rather to wallow and consume ribbed mussels[55,56].

Here, we show, through a series of field experiments, landscape-scale surveys, and models that, when hogs invade a salt marsh, a keystone mutualistic interaction between marsh plants and mussels is functionally lost, leading to mostly defaunated marshes that are more fragmented and recover more slowly from large-scale disturbances. Non-trophic mutualisms, as found in this marsh plant-mussel interaction, underlie the resilience of many ecosystems facing increasing global change. Our study shows that, when species like the destructive feral hog invade novel habitats, the ability for positive interactions to remain strong and provide crucial resilience is severely compromised.

## Results

**Hog effects on marsh fragmentation and recovery: 2-year patch recovery hog exclusion experiment.** We test the causal hypothesis that hog disturbance stalls the recovery and outgrowth of salt marsh grasses into surrounding mudflats with an exclusion experiment on the edge of patches of marsh grass recovering from disturbance in the Sapelo Island National Estuarine Research Reserve (SINERR). This 2-year hog exclusion experiment in two recovering Georgia marshes reveals that hogs hinder the ability for cordgrass to recover from common disturbances (e.g., drought, wrack deposition, and consumer disturbance) by slowing the rate at which plants revegetate mudflats over time, and thus help maintain fragmentation in salt marshes. Patch edges where hogs were excluded recover more over time than when hogs were present (Fig. 1, Supplementary Table 2; Time*Exclusion treatment: $F_{1,148} = 26.35$, $p = 0.001$). Cordgrass inside of exclusion cages recovers 3× faster (17.85 ± 1.3% vs. 6.85 ± 1.8% recovery per year) because hogs constantly trample the clonal plants that recolonize bare mudflats (see Supplementary Fig. 2 for image).

**Hog effects on salt marsh resilience and positive interactions: hog exclusion x mussel addition experiment.** To test the hypothesis that hog activity in marshes alters the strength of the positive effects of mussels on cordgrass and other associated organisms, we conducted a 3-year two-factor experiment manipulating the presence of both mussels and hogs using exclusion cages. When hogs are absent, mussels and cordgrass facilitate each other and increase bioturbating, herbivorous, and predatory burrowing crab densities (*Uca spp. Sesarma*

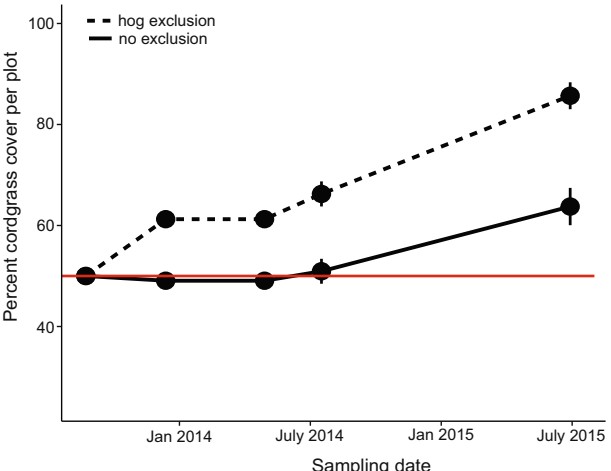

**Fig. 1 Experimental hog exclusions increase cordgrass patch recovery: patch recovery (mean % cover and standard error of live cordgrass per plot) in both uncaged control (solid line) and hog exclusion (dotted line) plots overtime at two different sites in the Sapelo Island NERR (n = 16 independent plots).** Percent cover started at 50% in each plot at the beginning of the experiment (red line) in June 2013 but plots with cages (see Supplementary Fig. 2) recovered much more thoroughly than uncaged controls that were trampled by hogs.

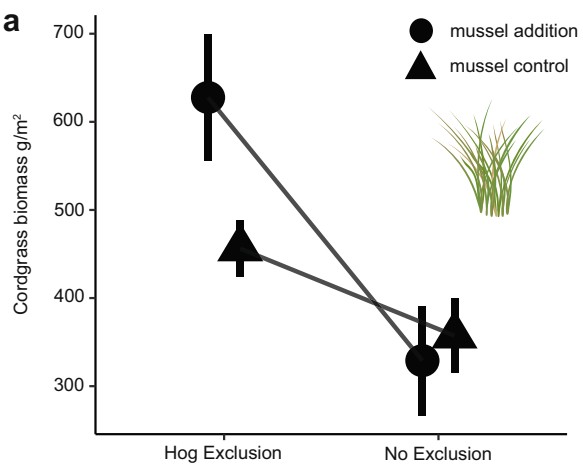

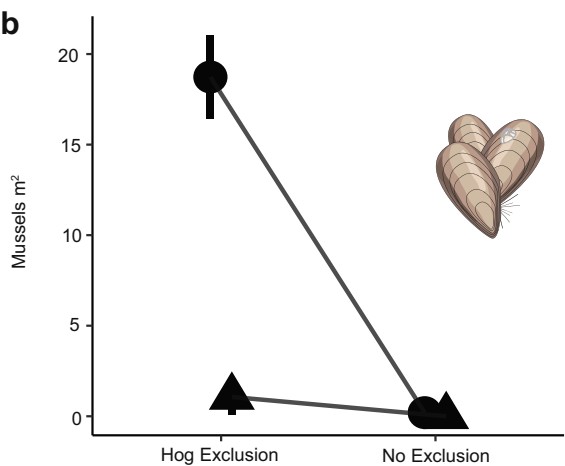

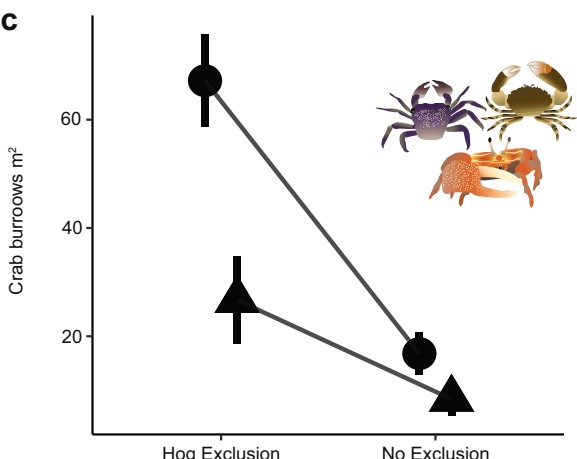

*reticulatum, Panopeus herbstii, Eurytium spp.*)[24]. Similar to findings in other experimental work[40,51], mussel additions increase cordgrass biomass by 1.5× (Fig. 2a), mussel survival is high (Fig. 2b), and crab densities are tripled (Fig. 2c) compared to our no mussel plots.

In uncaged plots, hogs completely disable the positive plant-mussel interactions: hogs reduce plant biomass by 48% in mussel addition treatments compared to the caged mussel addition plots where positive interactions are intact (estimate: −298.7 g cordgrass/m² difference; Fig. 2a, Supplementary Table 3; Hog*Mussels: $F_{1,35} = 5.86$, $p = 0.03$). The uncaged mussel addition plots were frequently disturbed and trampled, creating a final plant biomass equivalent to or slightly lower than uncaged plots with no mussels (estimate: −98.9 g cordgrass/m², $t = 2.8$, $p = 0.37$). Mean ribbed mussel density completely collapsed in these uncaged plots as only mussels inside of exclusion cages survive (18.7 vs. 0 mussels/m²; Fig. 2b; Hogs*Mussels: $F_{1,35} = 48.2$, $p < 0.01$). In addition, a mussel transplant experiment confirmed that mussels in cordgrass-free areas survive longer in the presence of hogs than in cordgrass patches (Supplementary Fig. 3a) because hogs either are unable to walk on marsh mud not stabilized by cordgrass roots or preferentially select more stable sediment for foraging. Importantly, cordgrass not associated with mussels is safe from hog trampling and uprooting (Fig. 2a), likely because cordgrass leaves and roots are low in nutrients and relatively unpalatable[56]. The effect of hogs on the cordgrass-mussel mutualism ripples through to the rest of the community, as hog presence negates the positive effect of mussels on crab burrow density by three-fold, from 32.6 ± 2.9 to 13.3 ± 1.5 crab burrows/m² (Fig. 2c, Hog*Mussels: $F_{1,3} = 7.58$, $p = 0.07$), a combination of direct consumption of crabs by hogs and from hog destruction of preferred burrowing habitat.

**Hog effects on fragmentation and recovery across marsh landscapes: drone survey.** To determine the relationship between hog activity and salt marsh fragmentation (i.e., the structure of remnant cordgrass patches), we conducted a drone survey recording the number and size of cordgrass patches in 14 marshes in Georgia and Florida that varied in hog activity (Supplementary Fig. 5). At large scales of up to 2 km², our drone

surveys show that hog activity has a strong, positive relationship with increasing fragmentation in salt marshes of the southeastern U.S.. Marshes with high hog activity (see Methods and ref. [57]), like the Matanzas State Forest, FL displayed significantly different patch recovery patterns than marshes with low hog activity like Faver–Dykes State Park, FL (Fig. 3). At higher levels of hog invasion, marshes contain more remnant patches (Fig. 4a,

**Fig. 2 Hog activity dismantles cordgrass-ribbed mussel positive interactions.** Effect of experimental hog exclusion and mussel addition on **a** mean cordgrass biomass, **b** mean ribbed mussel density, and **c** mean bioturbating crab burrow density at the end of the 3-year experiment. The interaction between hogs and mussels significantly affected all response variables. When hogs were allowed into plots, all positive effects of mussels on cordgrass (**a**) and crab (**c**) burrow densities were disabled. Triangles represent mussel control plots (no mussels added) and circles represent mussel addition plots (four mounds of 20 mussels added to each plot) (n = 40 independent plots across two sites), and error bars represent standard error.

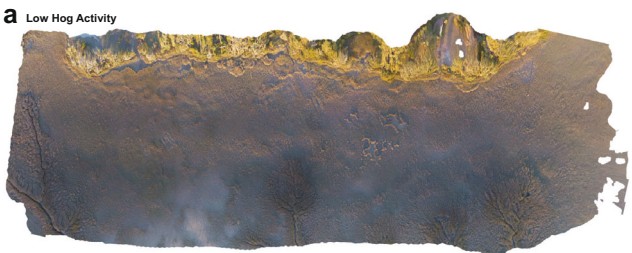

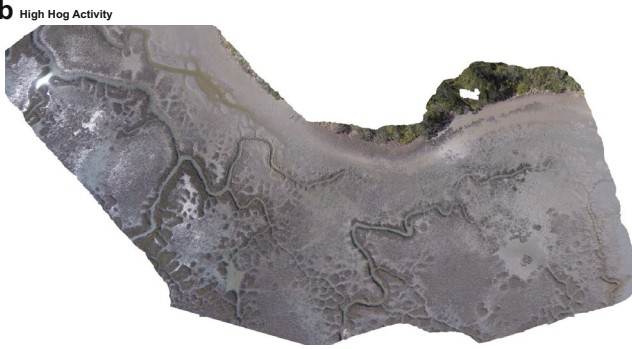

**Fig. 3 Drone imagery of hog effects across marsh landscapes.** Aerial imagery from two marshes, one with low levels of hog activity (**a**—Faver-Dykes State Park, FL), and one with high levels of hog activity (**b**—Matanzas National Forest, FL) showing examples of common patterns in patchiness in hog accessed marshes. In low hog activity marshes, mudflats tend to be smaller and patches are large but in high hog activity marshes, many small patches of remnant cordgrass are common inside of large mudflats.

Supplementary Table 4; high hog activity = 75.7 ± 20.4 patches/marsh, low hog activity = 16.0 ± 24.9 patches/marsh; mean patch number per marsh: $X^2 = 15.292$, df = 2, $p = 0.0001$), and, on average, these patches are much smaller (Fig. 4b, Supplementary Table 4b; high hog activity = 13.9 ± 2.7 m², low hog activity = 47.9 ± 7.2 m²; mean patch size: $X^2 = 10.24$, df = 2, $p = 0.005$). Our findings support our patch-level experimental results that find a positive relationship between hog activity and slow recovery times (Fig. 1).

**Hog effects on salt marsh resilience and positive interactions: mussel distribution survey**. Our survey of hog scat indicated that mussels are a focal food source for hogs, as 82.1% of surveyed hog feces (n = 190) contained mussel shell fragments (Supplementary Table 1). To quantify how local-scale hog disturbance correlates with patterns in ribbed mussel density and distribution, we conducted on-the-ground mussel density surveys along creekbanks and in the marsh platform at six marsh sites (0.4–0.8 km²) spread over 3.5 total km² of marshes in the Sapelo Island NERR in 2018. We found a negative correlation between hogs and mussel abundances

(total mussel abundance: Hog Presence*Location within Marsh: $X^2 = 46.7$, df = 1, $p < 0.0001$), implying that the dramatic reduction in mussel–cordgrass association shown in our experiment occurs at whole marsh scales. These results, combined with past surveys of southeast U.S. marshes that show mussel mounds typically cover 0.1–12% of the total marsh surface[24,48] and our scat survey, suggest that hogs preferentially feed in marsh areas with ribbed mussels. Hog-accessed marshes on Sapelo Island, GA have an order of magnitude lower mussel abundance than no-hog marshes (141.6 ± 30.4 vs. 2427.4 ± 145.1 mussels/marsh; Fig. 4c, Supplementary Table 5, Supplementary Table 4d). In addition, >99% of mussels are found in aggregations associated with cordgrass in hog-free marshes while the proportion of mussels that are associated with cordgrass in hog-accessed marshes is nearly halved (Fig. 4d, Supplementary Table 4e; Hog Presence: $X^2 = 28.6$, df = 1, $p = 0.03$, n = 50 patches sampled per marsh). This finding, plus the dramatic increase in the proportion of mussels on destroyed or defunct mounds (from 0 to ~78%), the decrease in the total percent cover of mussels per transect (from 7.2 to .47% along creek heads, 4.5 to 0.4% on the marsh platform), the reduction of mean mussel mound area (from 9.76 to 2.1 m² on the marsh platform) and increase in a number of singleton mussels (from 3 to 96 mussels found outside of a mound along with creek heads), indicate that most mussels in hog-invaded marshes were solitary, in small aggregations, or living without marsh grass, an extremely rare mussel distribution pattern in undisturbed marshes across the southeastern U.S.[24,58]

**Modeling long-term salt marsh resilience and recovery with hogs: marsh recovery model**. By examining how per capita predation pressure based on diet and activity in the marsh (i.e., hog behavior) affects marsh resilience in a spatially explicit model of the interactions between hog presence, mussel density, and marsh recovery rate, we find that hog predation on mussels can significantly impede the time needed for full marsh recovery following disturbance (Fig. 5). If mussels are a primary food source for hogs- as we have shown in our empirical work, then specific predation rates increase dramatically as hogs focus heavily on the remaining mussels when they search the marsh (Fig. 5a), a pattern we observed in our mussel transplant experiment (Supplementary Fig 3b). Here, a low abundance of hogs that completely focus on mussels can still deplete mussel densities despite low search efficiency at low mussel numbers (Fig. 5b). This increases marsh recovery times manifold as grass regrowth depends strongly on now depleted mussel mounds or recolonization (Fig. 5c). When mussels are a focus of hog foraging, post-die-off recovery is nearly impossible as hog presence must be lowered before recovery can occur. If mussels are just a secondary element of the hog diet (i.e., incomplete focus on mussels), mussel populations can remain high enough to fuel recovery, but marsh recovery time can still be long at elevated hog densities, and marshes under both scenarios will likely enter a state of unstable equilibrium (dashed lines). If no hog focusing on mussels occurs, for instance, because there are many alternative food sources, mussel density, and vegetation can still be reduced through trampling effects but it is unlikely that a decrease of mussel density in the area will lead to increased predation on remaining mussels. In this case, the remaining mussels will remain dense enough to fuel marsh recovery. When mussels are a focal element of the hog's diet, mussel cover, and hence marsh resilience can not easily be restored by hog culling. Our analysis highlights the key importance of hog behavior in terms of sustained predation effort in the face of reduced mussel numbers for the resilience of the salt marsh. Moreover, our data emphasize that the hogs will sustain predation effort despite decreasing prey number, leading to increased predation rates per mussel. This will

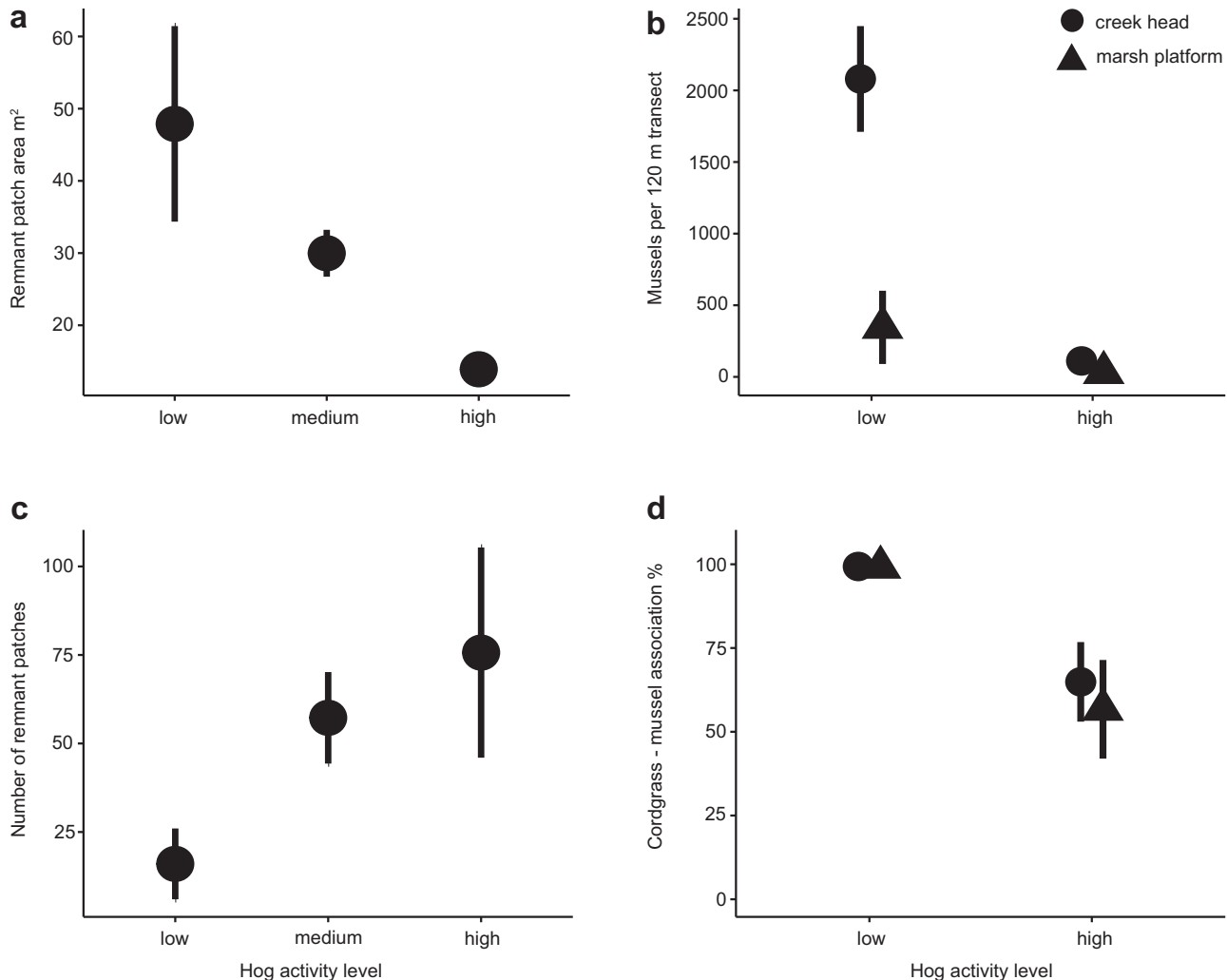

**Fig. 4 Hog activity increases fragmentation and disables positive interactions across marsh landscapes.** Analysis of drone imagery from marshes in Georgia and Florida ($n = 14$) revealed that (**a**), mean area of remnant patches ($m^2$), and (**c**) mean number of remnant cordgrass patches both increase with increasing levels of hog activity. Distinct marsh patchiness (Fig. 3) is the result of this strong correlation between higher hog activity and increased, more permanent disturbance Whole marsh ribbed mussel surveys in the Sapelo Island NERR that found (**b**) mean ribbed mussel density was much lower along creek banks (circles) and in the marsh platform (triangles) in hog-accessed marshes. In addition, the frequency of association between ribbed mussels and cordgrass (mean % association, **d**) in both parts of the marsh drastically declined from nearly 99% in hog-free marshes to around 60% in hog-accessed marshes. Error bars represent standard error.

significantly impair the intrinsic resilience of the salt marsh, which will then entirely depend on recolonization from outside of the die-off area (as in ref. [24]).

## Discussion

Results from this study reveal that a large invasive consumer significantly impairs the positive species interactions that form the basis of resilience in southeastern U.S. salt marshes. We draw this conclusion based on four lines of evidence across different spatial scales. First, multi-year exclusion experiments show that hogs significantly reduce post-disturbance (i.e., drought, Supplementary Fig. 4) cordgrass recolonization rate of mudflats, simply from trampling as they travel through the marsh (Fig. 1). Second, this activity appears to cause differences in fragmentation detected from drone surveys across Florida and Georgia marsh landscapes. As hog activity levels increase, the marsh landscape is covered with many small recovering patches of cordgrass (Figs. 3, 4a, b), a configuration that drastically slows total marsh recovery time, lowers biodiversity, and reduces multifunctionality[49,58].

Next, our manipulative hog and ribbed mussel field experiment demonstrates mechanistically that the key animal–plant interaction in salt marshes is strongly affected by the presence or absence of hogs. When hogs are excluded, the interactions between cordgrass, mussels, and marsh crabs are positive, facilitating plant growth and normal recovery. When hogs are present, these positive interactions are completely lost as hog foraging activity inhibits plant growth (Fig. 2a) and reduces the abundance of mussels (Fig. 2b) and other bio-regulating marsh species (i.e., crabs, Fig. 2c). Thus, hogs destroy the marsh facilitation cascade en route to mussel consumption. Hog-targeted destruction of the cordgrass-mussel mutualism is general across marshes as the association between cordgrass and ribbed mussels that facilitates recovery breaks down when hogs are present (Fig. 4c, d, Supplementary Table 5). Lastly, our marsh recovery model suggests that, when hogs specifically target ribbed mussel mounds, future recovery ability is impaired and reduced, as recovery from large-scale die-off may never occur before mussels can repopulate an area (Fig. 5). This result highlights that hog-induced losses of mussels can significantly weaken the resilience of southeastern

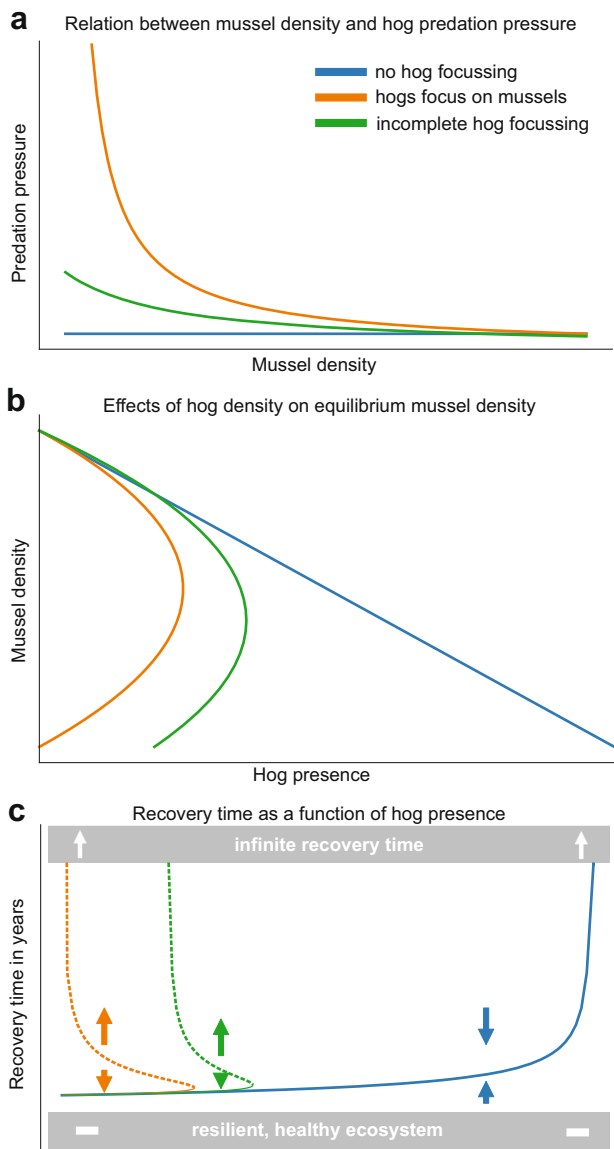

**Fig. 5 Hog behavior affects long-term marsh resilience and recovery.**
Results from a salt marsh recovery model show (**a**) how differences in predation pressure from three different hog foraging behaviors deplete mussel densities at different rates and (**b**) how increasing hog densities reduce mussel densities at different rates based on foraging behavior. Hog foraging behaviors include no focus of hogs on mussels (i.e., haphazard consumption or trampling when hogs enter the marsh, blue line), the incomplete focus of hogs on mussels (i.e., targeted consumption with decreasing predation pressure as mussel densities are depleted, green line), and complete hog focusing on mussels (i.e., targeted consumption with increasing predation pressure as mussel densities are depleted, orange line). These patterns generate significantly different cordgrass recovery times **c** where hog's focus intensity on mussels drives time until whole marsh recovery, as ribbed mussel density and configuration indirectly increases marsh recovery. Dashed lines represent unstable equilibria where, if pushed past, marsh recovery time could be infinite.

U.S. salt marshes and thereby reduce their protective effect on coastal communities[59].

Degradation of facilitation cascades, habitat cascades and mutualisms will have long-lasting effects on the resilience of important coastal ecosystems like marshes, mangroves, seagrasses, kelp forests, and coral reefs[60,61], by lowering coastal defense in the

areas that shield people and property from sea-level rise and storms[44]. Here, in an ecosystem under threat from climate extremes, the feral hog invasion lowers the survival of both primary and secondary foundation species (Fig. 2a, b) and alters community structure (Fig. 2c), to reduce long-term resilience and slows large-scale recovery. This degradation of positive interactions across space and time has direct implications for coastal defense because promoting these interactions, especially in foundation species, is a crucial element for coastal ecosystems to persist and protect[59,62]. Thus, continued mismanagement of invasives that destroy positive interactions and foundation species will affect the coast in more extreme ways because the sea-level rise and drought effects, for example, are greatly amplified by defaunated habitats[31,63,64].

In many ecosystems powered by positive interactions, the effects of species introductions on keystone facilitation may be stronger and more wide-reaching than the more well-known invasion effects on habitat structure or biodiversity (e.g.,[5,9]). When hogs invade a salt marsh, resilience is reduced because the otherwise positive cordgrass and mussels association now impose a negative effect on the survival of both species, as cordgrass survives without mussels (Fig 2a) and mussels survive hog invasions in bare areas where cordgrass does not stabilize the sediment (Supplementary Fig. 3a). Similar to severing key trophic interactions, the effects of interaction reversals in other ecosystems can cascade throughout the entire community, impairing biodiversity, multifunctionality, and resilience[24,58]. For example, in Guam, the invasive brown tree snake preys on seed-dispersing birds, reducing plant productivity across large scales[17,65]. In the Caribbean coral reefs, the invasive lionfish (*Pterois volitans*) can reduce the density and recruitment of many herbivorous fishes that keep coral free of macroalgae[66,67]. In Tasmanian kelp forests, climate-induced range expansion of urchins and subsequent overgrazing has caused key kelp positive interactions to collapse, reducing community biodiversity and ecosystem functioning on large spatial scales[68–70]. Determining when and where invasives may alter positive interactions and, in turn, evaluating how resilient these positive interactions are to change, is crucial to prevent the collapse of systems supported by facilitation[60].

Our study demonstrates that, when large consumers enter novel habitats through invasions, range expansions, or reintroductions, direct and indirect effects on resilience and recovery will be relevant over long-time scales and large spatial scales. Hogs compromise the future ability of southeastern U.S. salt marshes to adapt to climate change through trampling the edges of recovering patches (i.e., direct disturbance effect; Fig. 1) and targeting ribbed mussel beds to reduce marsh-scale mussel density by an order of magnitude (i.e., indirect effect on mutualism; Figs. 4c and 2b). These effects will have long-lasting effects on coastal marshes as mussel loss alone can add decades to the drought-recovery process by degrading ecosystem functions that fuel recovery[40,57,71]. Because hog activity both directly and indirectly affects marsh recovery, regrowth of plants in hog-invaded marshes should occur much slower than in Louisiana marshes denuded by invasive nutria (*Myocastor coypus*). Nutria can consume hectares of vegetation but, because they do not consume ribbed mussels, nutria-used marshes may be able to recover within a just year post-eradication[72]. Hog invasion of marshes is thus more similar to predation by introduced species-feral cats and red fox (*Vulpes vulpes*)—on Australia's digging mammal species that generates impoverished landscapes with slow rates of seed germination and a severely limited ability to recover from changed fire regimes[73,74]. Similarly, introduced Artic foxes (*Alopex lagopus*) predation on seabirds lowers nutrient input and soil fertility, creating an alternate stable state as grasslands transform to dwarf shrub/forb dominated ecosystems[75]. Thus, invasive consumer behavior can cause

important changes in ecosystem resilience over space and time, amplified by the degradation of species interactions that support ecosystem functioning.

Highlighted in our marsh recovery model, hog foraging behavior can affect marsh resilience in different ways. When hogs do not focus predation on mussels, their impacts are limited to just trampling, a linear relationship that increases directly with hog abundance (Fig. 5a– blue line). Trampling effects may not affect marsh resilience at low hog abundances (Fig. 5c–blue line) but focused predation by hogs (Fig. 5a–orange line) can lead to a positive feedback loop where increased focus on mussels increases the predation rate on the remaining mussels and leads to reduced mussel cover (Fig. 5b). The loop repeats again with an increased focus on the remaining mussels, a general process that is suggested to be relevant beyond the salt marsh (e.g., semi-arid systems[76]). This foraging behavior will lead to a potentially permanent collapse of the mussel population, severely reducing the intrinsic recovery potential of the salt marsh following drought events, which will entirely depend on recolonization from outside of the die-off area[24]. Mussel-driven recovery would then require a pause in hog activity and a corresponding recolonization of bare marsh, which is difficult because mussels strongly prefer to settle in conspecific aggregations. Mussels can recruit to areas that are more difficult for hogs to access (i.e., creekbanks and soft sediment) so the permanence of recovery and resilience loss depends on both recolonization and hog activity across the marsh landscape[77]. The spatial and temporal scale of hog effects can be predicted by these foraging differences combined with large-scale patterns in hog habitat usage, where the most intense hog foraging activity is found where the salt marsh is surrounded by expansive hardwood forests[57]. Marsh disturbances are expected to increase in frequency[46] and the spread of hogs into new coastal habitats (e.g., dunes and mangroves) will continue, especially as management beyond intense hunting pressure is unreliable and/or unsuccessful[54,78–82]. Thus, as hogs access more areas throughout the coastal U.S., valuable habitats will be more fragmented and lower in elevation,[83] creating areas that are slow to recover to disturbance and less resilient to sea-level rise. Hog-invaded marshes will also have lower levels of ecosystem functioning since marsh multifunctionality can be controlled by a few, superabundant species like ribbed mussels and burrowing crabs[24,84] that all decline in the presence of hogs (Fig. 2b, c).

Coastal ecosystems are increasingly affected by abiotic and biotic forcings like drought, habitat fragmentation, and species invasions that simultaneously apply pressure to the positive interactions that drive resilience. Our experiments, surveys, and mathematical models suggest that southeastern U.S. marsh resilience will be especially vulnerable to these effects without intense feral hog hunting or the reintroduction/rewilding of natural predators[85]. Large invasive consumers like hogs represent management challenges in increasingly novel human-influenced ecosystems[86]. Forecasting how habitat resilience will change with invasive species requires an increased understanding of the wide-reaching effects of altered interactions cascading throughout food webs[9]. Current predictions of ecological responses to global change stressors cannot assume interaction networks in the future will function in the same way as the present. Our work shows that the common invasive effects— outcompeting, consuming, or otherwise reducing the abundance of native species—are compounded by indirect modification of key positive interactions that determine how ecosystems respond to stressors. As managers attempt to amplify positive interactions to achieve conservation goals[61], anthropogenic activities will continue to shift baselines in the species interactions that support ecosystem function and services.

## Methods

**Hog effects on marsh fragmentation and recovery: 2-year patch recovery hog exclusion experiment**. In May 2013, we selected two marshes within the Sapelo Island NERR with observed hog activity, Kenan Field and Miller Pump (~4 km away from each other). We marked 16 replicated 2 × 2 m plots on the edge of similarly sized recovering marsh patches (between 9 and 16 m²) within a bare mud-remnant patch matrix, where no two patches were closer than 5 m from each other. Each plot began the experiment with the same initial starting percent cover (50%), in order to properly capture recolonization and recovery rate. Here, we performed a hog exclusion experiment ($n = 8$ hog wire exclusion cages, 8 control open plots per site, Supplementary Fig. 2) during marsh revegetation after years of severe drought (Supplementary Fig. 4). The mesh size of the hog wire was 20 cm² and cages were 1.5 m tall, allowing access to all marsh species including other large mobile predators like blue crabs, fish, and raccoons. We found evidence (e.g., raccoon tracks and crushed mussel shells) that these other predators were allowed equal access to caged and uncaged plots. The only other potential haphazard exclusions were deer, horses, or cows, which are not known to consume marsh organisms and only occur at very low densities in our study sites[87]. In this study, we observed no buildup of wrack material, hog exclusions with smaller mesh (2 cm²) have shown no evidence of flow artifacts and mesh size was wide enough that shading effects were minimal.

For each plot, we standardized initial plant density and percent cover (130–150 cordgrass stems/m²; 50% cover) and selected plots with nearly identical initial invertebrate densities (5–10 fiddler crab Uca sp. burrows/m², 0–25 snails Littoraria irrorata/m², and 0–10 ribbed mussel Geukensia demissa/m²) to ensure there were no differences between plots and within sites. Initial plant cover was not manipulated at the beginning of the experiment, rather, we explicitly marked plots so that each plot began at 50% cover, with half of the plot covering the vegetated patch and the other half covering unvegetated mudflat, i.e., where cordgrass recovery would occur (see Supplementary Fig. 2). Over the course of the experiment, we recorded the percent cordgrass cover using a gridded quadrat. Final data on this experiment were recorded in July 2015 after 24 months in the field. We analyzed the effect of hog exclusion on plot recovery (% cover) using a linear mixed-effects model fit by maximum likelihood with the site and plot number as a random effect to account for repeated sampling, and sample interval (i.e., time) and exclusion treatment as interacting fixed effects. We evaluated randomized quantile residuals to evaluate assumptions using the DHARMa package (v0.4.3)[88] and found that assumptions were met. All analyses were conducted in R (v4.1.0)[89] using lme4 (v1.1–27.1)[90] and lmerTest (v3.1–3)[91].

**Hog effects on salt marsh resilience and positive interactions: hog exclusion × mussel addition experiment**. We conducted two experiments to determine how hogs affect resilience-enhancing positive interactions in salt marshes. First, to determine whether cordgrass mediates hog predation intensity on ribbed mussels, we conducted a mussel transplant experiment at Kenan Field and Miller Pump marsh (see Supplementary materials, Supplementary Fig. 3) and found that hog predation rate on mussels transplanted into cordgrass patches is higher than on mussels transplanted into unvegetated mud. Second, to test the hypothesis that hogs alter the strength of the positive effects of mussels on cordgrass and other associated marsh organisms, we conducted a 3-year two-factor experiment manipulating the presence of both mussels and hogs using exclusion cages. At both sites, we selected 20, 2 × 2m plots in cordgrass patches and randomly assigned each to one of the following treatments: (1) Hog exclusion, no mussels, (2) Hog exclusion, mussels added, (3) Hog control, no mussels, and (4) Hog control, mussels added. Mussel addition treatments consisted of four separate mussel mounds with 20 individual adult mussels per mound, reflecting natural mound densities for coastal Georgia marshes[40]. Mussel addition treatments were reapplied at the end of year 1 (November 2013) and year 2 (November 2014) of the experiment as some (~2–6 mussels/cage/year) mussels inside of hog exclusion cages were consumed by raccoons or experienced mortality from transplant stress. Throughout the course of the experiment, we measured live and dead grass mass, as well as marsh community structure that has been shown to be positively affected by this mutualism (i.e., fiddler crab and mud crab burrow density). The experiment ended in December 2015. We analyzed the effect of cage treatment and mussel treatment using a linear mixed-effects model with the site as a random effect and hog exclusion and mussel addition treatments as interacting fixed effects. Models were checked for and met assumptions as above.

**Hog effects on fragmentation and recovery across marsh landscapes: drone survey**. Sites for our drone surveys were chosen from 14 marshes in Georgia and Florida known to vary in hog activity, described in a recently published method that uses spatial land cover information and on-the-ground hog impact surveys to predict the location and extent of feral hog activity in salt marshes[57]. Briefly, low, medium, and high levels of hog activity used in our analyses and figures refer to the total predicted area of hog disturbance within a 10,000 m² area (approximately 500 m long and 20 m wide) of the marsh (low = 95 m² of disturbance, medium = 1201 m², high = 2105 m²). Our chosen sites were a subset of the hog-activity classified sites described in ref. [57], chosen within Georgia and Florida to maximize variation in estimated hog activity and minimize variability in the history of drought and habitat modification across large scales, then standardized for size

(between 0.6 and 1 km²). We were not able to control for distance to human populations in this survey, but no sites were within 1 km of heavy human development. We selected marshes with full cordgrass monocultures (i.e., short-form cordgrass in marsh platforms and tall form cordgrass along creekbanks) and similar creek sizes in order to minimize variation in elevation, salinity, inundation time, mussel recruitment, and percent organic matter. Using a DJI Phantom 3 drone with a GoPro Session camera attached, we surveyed 28 15,000 m² areas (approximately 50 m wide, 300 m long), at least 500 m apart, along the Atlantic coast of Florida and Georgia. We conducted two surveys/flights at each of 14 sites: 5 high activity marshes, 6 medium activity marshes, and 3 low activity marshes. Drone path length, which determines the area covered, as well as the height of the drone, was standardized (30 m high) to control the total area covered for each drone flight. We created a photo mosaic of each flight using PhotoscanPro (v1.2)[92]. We calculated a total number of live cordgrass patches and mean patch size for each of the flights, scaling each mosaic with a known, marked area on the ground, and manually circling and measuring patches using ImageJ (v1.48)[93].

We analyzed the effect of hog activity on patch number, patch area, and mud cover area using generalized linear mixed models fit by using maximum likelihood with hog activity as a fixed effect and site as a random effect for each response variable. Patch number (a count variable) was fit with a Poisson error structure, while mean patch area and mud area were fit with a gamma error structure. We evaluated hog activity using likelihood ratio tests.

### Hog effects on salt marsh resilience and positive interactions: mussel distribution survey.
Mussel abundance surveys were conducted at three known hog-accessed and three hog-free sites in the Sapelo Island NERR. Sites were selected based on proximity to human development, a proxy for hog-access on Sapelo Island (Supplementary Fig. 5, Georgia Department of Natural Resources *pers. comm*, M. Hensel *pers. obvs*) and were selected due to similarities in elevation, creek size and number, and total marsh area, all key components of mussel density across marsh landscapes[77]. Within each marsh, we selected 120 m² transects in each of two area types: the creek head, where water first enters the marsh platform and mussel cover is highest[94], and the high marsh platform, zones of short-form cordgrass located 30 m from the terrestrial border and >50 m from the nearest creek head. Within each 120 m² transect, we counted all singleton mussels and mussel aggregations and measured the dimensions of each mussel and mound encountered ($L \times W \times H$). For each singleton mussel and mound, we identified whether the mussel or aggregation of mussels was associated with cordgrass through attachment by byssal threads, and categorized the local area type as marsh platform, active mussel mound, or defunct mussel mound—locations where mussel mound remnants are visible (i.e., shell fragments, bump in primary productivity, and mussel pseudofeces buildup), but no mussels or very few remained.

We analyzed the correlation between hog-access and within-marsh location on the total number of mussels using a generalized linear mixed model fit by maximum likelihood with a negative binomial error and logit link, with hog access-level (high or low) and marsh area (creek head or marsh platform) as interacting fixed effects and site as a random effect. We corrected *p*-values of post hoc comparisons using Tukey's correction method with emmeans (v1.6.2-1)[95]. We evaluated randomized quantile residuals as a test of assumptions as before and found no violations. To analyze the correlation between hog-access and within-marsh location on the percent association between mussels and cordgrass, we used the same generalized linear mixed model fit by maximum likelihood with a binomial error.

### Modeling long-term salt marsh resilience and recovery with hogs: marsh recovery model.
To test for the long-term implications of hog trampling and mussel predation, we integrated hog predation of mussels within an existing numerical model of the relationship between mussel density, cordgrass growth, and salt marsh recovery[96] (see Supplementary Materials for more model details). We extended this model to include a mussel population, whose cover can vary between zero and a maximum standing crop (defined as one) which is equivalent to the maximal mussel cover observed in the area of about 10%. We presumed a set hog population, which exerts a certain predation pressure on the mussels, determining mussel cover. For simplicity, we presume that the cover of the vegetation is proportional to that of mussels[40,58], which follows a logistic growth rate. Modeled hog population predation pressure varied based on three different predation methods: no focus of hogs on mussels (i.e., haphazard consumption or trampling when hogs enter the marsh), the incomplete focus of hogs on mussels, and complete hog focusing on mussels. No focus of hogs on mussels approximates a trampling-only behavior, as few mussels are consumed and vegetation would be affected most at high hog densities. The incomplete focus of hogs on mussels describes a common prey depletion pattern where predation pressure on remaining mussels decreases as mussel densities decrease. Complete hog focusing on mussels describes the prey depletion pattern we observed in our mussel transplant surveys where predation pressure on remaining mussels was higher as mussel densities decreased (Supplementary Fig. 3b). Assuming a random distribution of mussel patches in the landscape, we then computed the time the vegetation needs to recolonize the landscape following a disturbance that removes all vegetation outside of the mussel patches, mimicking a drought-induced die-off event[24,97].

**Ethical compliance**. While we did not make any contact with our study species, the feral hog *Sus scrofa*, we did obtain IACUC permits for our exclusion experiments. At the genesis of this project, lead author Hensel and second author Silliman were at the University of Florida and received UF IACUC approval (201207684) on 11/5/2013. No contact with vertebrate animals occurred during the course of this experiment and we observed no entanglement of any animals in our cages.

**Reporting summary**. Further information on research design is available in the Nature Research Reporting Summary linked to this article.

## Data availability
Datasets that support the analyses and figures of this study have been deposited in the public repository: https://github.com/mhensel/HogsInTheMarsh.

## Code availability
Code to reproduce the analyses and figures of this study are stored in the public repository: https://github.com/mhensel/HogsInTheMarsh. https://doi.org/10.5281/zenodo.5225047.

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

## Acknowledgements

We thank the staff at the Sapelo Island NERR for logistical assistance during this project, as well as teachers from the LTER Schoolyard Program for assistance in the field. We also thank Christine Angelini, Jen Bowen, Jon Grabowski, and Ron Etter for constructive feedback on earlier drafts. This research was funded through an NOAA-NERR Graduate Research Fellowship awarded to M.J.S.H. (Award no. NA12NOS4200087). This work is a product of the Georgia Coastal Ecosystems LTER project.

## Author contributions

M.J.S.H., B.R.S., and J.E.K.B. conceived of the idea; M.J.S.H., S.J.S., S.M.C., J.vd.K., and E.H. collected the data; M.J.S.H. and J.vd.K. analyzed the data; M.J.S.H. drafted the paper with substantial input from all authors.

## Competing interests

The authors declare no competing interests.
