## [Peer Review File · Nature Communications]

Reviewer #1 (Remarks to the Author):

The manuscript by Hensel et al. provides a very interesting example of the ways in which non-native invaders can disrupt community processes by degrading or eliminating positive interactions among key species that contribute to resilience. The study brings into line manipulative experimental studies with observations on larger spatial and temporal scales and a predictive model of recovery rates. The issue is an important one since the ability of systems like this, which are physiologically stressful to start with, to recover following anthropogenic disturbances like invasions has lessons for many other systems. In short, the authors make a strong case for introduced hogs dramatically reducing cover of cordgrass and densities of mussels. They also show that this results in the inability of either cordgrass or mussels to facilitate the other after hog related reductions.

I think the case for hogs disabling positive interactions is a good bit stronger than the case for hogs reversing the direction of the interactions or in the authors words 'flipping' the interaction. Here is where I think the authors overstate their case a bit, and I don't feel they need to do this to have an important story. I really tried to look at Fig 4A to see a reversal in the effects of mussels on cordgrass. What seems clear to me is that cordgrass biomass is substantially lower in 'Hog Control' areas. However, cordgrass abundance looks about the same for 'Mussel Addition' vs. 'Mussel Control' in those areas. Yes, ever so slightly lower with mussels, but not significantly so, and the authors should avoid making statements like "...a final plant biomass lower than unmanipulated plots" when the statistical result is NS ($p = 0.37$). Thus, I see no evidence of hogs 'flipping' the interaction. The authors also make the point in lines 156-158 that mussels in cordgrass-free areas survive longer in the presence of hogs than in cordgrass patches (and Supp Fig 3) because of the inability or unwillingness of hogs to forage in unstable sediments, it's still based on situations where both mussels and cordgrass are at low abundance, so these are not strong negative interactions. I would recommend rather than trying to make a lot of a weak case for actual reversal of positive interactions to highlight the very clear case for the complete loss of positive interactions. The compounding effect of hog predation is really to decouple this important interaction which clearly results in delayed is not complete loss of recovery. My overall conclusion is that this is a carefully conducted study with a really interesting and important message for both fundamental science and the management of salt marsh systems.

A few additional minor areas that should be amended. One is the authors should be a bit more cautious about the conditions under which hog predation will lead to the complete collapse of the mussel population. On lines 265-267, they state "This will lead to collapse of the mussel population, severely reducing the intrinsic recovery potential of the salt marsh following drought events, which will entirely depend on recolonization from outside of the die-off area." But recolonization from outside is an important process that could substantially delay or prevent complete die-off. Unfortunately, their very simplistic model of mussel population dynamics reflects reproduction as a simple function of current population size without any description of external recruitment. Of course mussel larvae can travel distances much greater than the spatial scale of sites with and without hogs on Sapelo Island, which based on Supplemental Fig 5 look to be separated by 10 km or less. Admittedly there is some feedback between adult population size and subsequent recruitment, since local fecundity will matter and ribbed mussels do preferentially settle in conspecific aggregations. But the prospect for mussel recover may not be as dire unless the spatial scale of hog disturbance is a great deal larger.

A second issue is that I was surprised the authors never mentioned another very damaging introduced megaconsumer in the marshes of the southeastern US, the oversized rodent Nutria (*Myocastor coypus*), which has also dramatically reduced cordgrass abundance. Some mention and comparison with that invasion and the rates of recovery would be helpful, since presumably Nutria consume only cordgrass and not mussels. Thus, it would be predicted that the marshes would recover faster. Nutria has invaded large areas over decades (although controlled in many places now) so certainly there must be some useful recovery rate data to compare with this study.

Minor fixes

Please change the Y axis in Fig 2B to reflect a real density rather than 'per marsh'. If I read the Methods correctly, mussels were counted on 120 m transects, so possibly these marsh counts are per 120 sq m

Reverse the order of the entries in the legend of Fig. 3 so the order matches the lines (put Exclusion above Control). Also there is no need for colored lines (there are only two).

Lines 384-388 describe adding 20 mussels to each of four mounds within the 2 x 2 m areas, which would be 20 mussels per sq m. (approx.). So how were mussel densities in Fig. 4B as high as 40 per sq m in the mussel addition treatments without hogs?

Fig 4 needs to have letters A, B, C added.

Reviewer #2 (Remarks to the Author):

Summary:

The authors combine seven different studies (an observational study of landscape structure at 10 sites, two observational and three experimental studies at one location [Sapelo Island], and a population model) to explore the effects of the presence of invasive hogs on the relative abundance of ecologically important cord grass and mussel species in marshlands within the southeastern United States. Based on these varying lines of evidence, they conclude that the effect of hog predation on mussels and trampling on cordgrass is likely responsible for driving continued large-scale patterns of habitat degradation in marsh systems in the region.

The authors have generated a commendable amount of information on this topic among the various studies and data sets. In particular, the hog removal studies at Sapelo Island clearly show that hogs have a substantial effect on the recovery of degraded cordgrass habitat and that predation on mussels and likely trampling together clearly significantly reduces the presence and abundance of both species. However, I found that given the huge amount and variety of information sources, many key details of each study were missing from the manuscript (and supplement), and the various pieces were not weaved together in a convincing way to connect these local effects with large scale patterns of habitat change and resilience.

In particular, many key elements that are essential for convincing the reader that the estimation of hog 'activity' at landscape scales is robust were missing from the description of the landscape correlation analysis, and any other likely drivers of the history of fragmentation observed are not addressed (a limitation of only having 10 sampling units for statistical analysis), such as proximity to human settlements and thus other sources of disturbance. In addition, while linking both predation and trampling by hogs ecosystem recovery is an exciting idea with modelling potential, it does not appear that trampling is explicitly included in the model (only predation, and via three scenarios that aren't well linked to empirical data). It is not clear how mussel cover is linked to system recovery time in the paper- what is the metric of recovery used? Are the authors equating mussel increases with ecosystem recovery? If so, what evidence is there to suggest that this species alone indicates all the core functions of a recovered system?

I also find the figures to be missing key elements that are integral to threading these disparate approaches together, particularly regarding the large-scale correlation (location and selection of the sites, regional coverage and proximity to other drivers such as human settlements), and modeling results (what are the units? How do these figures relate back explicitly to the model structure described in the supplement?).

Beyond these links across scales, it seems that the main claim of the manuscript (that positive interactions turn into antagonistic ones as a result of invasion) hangs on the result shown in Figure

3A comparing cordgrass density between mussel addition and non-addition treatments in plots where hogs were present— reported as an ‘antagonistic’ interaction between mussels and cordgrass. The effect size here is very small and not significant in terms of the influence of mussels (Supp Table 5A), and occurring in the context of extremely low abundances of mussels (i.e. no difference between $\frac{3}{4}$ treatments in Fig 3B, save the addition + exclusion treatment, which also suggests that mussels are likely not playing a functional role in these treatments). Given the small effect size and low mussel density during this effect, how likely is this to be truly an antagonist interaction (and what would the mechanisms be) vs another effect associated with changing mussel densities over the course of the experiment (e.g. extra hog trampling in the mussel addition plots), which seems far more likely?

Should the authors wish to retain the integration of these seven hefty data sets—which could be published convincingly in several parts with sufficient details—I believe the piece is a better fit for a longer format publication where there is space to provide the necessary details and links between the myriad studies. Below are additional detailed comments that I hope the authors will find helpful in revising their piece.

Core feedback and suggestions:

Landscape structure correlation study: Where are the sites located? How many sites have been binned into each activity group? How do the sites (and data) relate to the cited Sharp and Angelini paper cited for the method? Are these the same data presented in this manuscript, or was the same method applied to new sites? If it is the latter, far more information needs to be provided beyond the brief description in the methods (with no additional information in the supplement). I could think of other disturbance factors that could also drive patch number and size in these areas (proximity to human populations, history of drought and habitat modification). Yet no covariates are presented in the analysis; however, with only 10 sampling units it is unlikely to be possible to examine additional drivers due to statistical limitations.

Hog fecal analysis: The fecal analysis survey results play a major role in the Results section of the manuscript text as evidence for the consumption of mussels in the region. However, the methods for this data collection (i.e. approach to sampling, number of samples, analytical techniques) are not included in the study Methods section, and instead given in part in the supplement only. It is also not clear how these data are used to inform the predation component of the modeling portion of the paper. The authors create three predation ‘scenarios’ (haphazard predation, incomplete focus on mussels, and complete focus on mussels), but its not clear what supports these scenarios and how they are linked back to evidence of predation relative to other prey (presumably from the scat analysis).

Mussel population dynamics: The idea of linking both predation and trampling to ecosystem recovery is an excellent one and question that could be well addressed through modeling. However, while the study states that the model links both the effects of hog trampling and predation on marsh recovery dynamics, it is not clear how trampling is incorporated into the model (no equations are provided, or translation between original model and the one used), and it is unclear how changes in mussel population size are translated into ‘marsh recovery’ specifically, which is only mentioned a few words at the end of the Methods (lines 410-411) and not in the supplement. The graph of results (Figure 5) contains no units, and it’s unclear what hog presence means.

Hog exclusion experiments: The hog exclusion experiment, and two-factor hog exclusion and mussel addition experiment are solid evidence for the effects of hogs via predation and trampling (though the effects of the two mechanisms can’t be disentangled in the latter experiment). To my mind, these are core elements of the study that should be given more focused attention on their own and could easily comprise a stand alone ecologically focused manuscript.

It seems that the main thrust of the manuscript hangs on the result show in Figure 3A (thought here are no panel labels provided) comparing cordgrass density between mussel addition and non-addition treatments in plots where hogs were present— reported as an ‘antagonistic’ interaction between mussels and cordgrass. The effect size here is incredibly small, and occurring in the

context of extremely low abundances of mussels (as seen in Figure 3B- no difference between $\frac{3}{4}$ treatments, save the addition + exclusion treatment, which suggests that mussels are not playing a functional role in these treatments). Given the small effect size, and low mussel density of this effect, how likely is this to be truly an antagonist interaction (and what would the mechanisms be) vs another effect associated with changing mussel densities over the course of the experiment (e.g. extra hog trampling in the mussel addition plots)?

Additional detailed feedback:

Suggest reviewing plots to standardize text font and size/style between figures as it's quite variable.

What are the measures of error in Figures 2 and 4?

Figure 1: It would be more effective to show the study region and all 10 sites and also highlighting the location of the local experiments at Sapelo Island in relation to the observational data collection. You could use two insets to demonstrate the extremes of the range.

Figures 2 and 4 need panel labels.

Figure 5: What does 'Hog presence' mean? What are the units and scales for all axes?

Supplemental model description: How is trampling considered in the model? How is the 'recovery time' response variable measured (i.e. Figure 5)?

Supplemental Table 1 and fecal sampling description: It's clear that hogs eat mussels, but what seems more important for linking to the modelling components is the composition of feces in each sample, rather than the ubiquity of any shell across fecal samples. How does the relative abundance (count, weight, volume) of shell compare to other diet items?

Reviewer #3 (Remarks to the Author):

This is an interesting manuscript that brings together multiple approaches across a range of scales to investigate the influence of invasive hogs on marsh recovery via their disruption of a positive facilitation. The use experiments, modeling, and landscape scale surveys to show that, when hogs are absent, mussels and cordgrass have a facilitative relationship that increases the likelihood and rate of marsh recovery after a disturbance (e.g., drought). However, in the presence of invasive hogs, this facilitation is reversed, and mussels attract hogs to forage in cordgrass clusters, which both reduces mussels and cordgrass, as well as the likelihood and rate of marsh recovery. Thus, the presence of hogs on the landscape translates into more bare mud ground and higher number of smaller patches of marsh grass. These findings have important implications for the conceptual understanding of facilitative relationships and potential impacts on invasive species, which has not been very well-studied in the literature. They also inform conservation of marsh ecosystems in the southeast by illustrating the imperative nature of hog removal for coastal resilience, which is an issue of increasing concern. Thus, I think this paper makes an important and novel contribution to the literature, and I expect this paper will be of high significance for several fields and will be well-cited. That said, there are several areas I think the manuscript could be strengthened that I think should be addressed prior to publication.

First, the multiple approaches and suites of data presented are a great strength of the paper, but I found them to be presented in a very confusing way. Particularly in a paper with this format, where methods come after results and may never be read, I would include a brief summary statement of what you did at the start of each results section. For example, Li 145: "We conducted a factorial experiment in which we added mussels and excluded hogs to examine their effects cordgrass and marsh consumer density." I would also suggest re-organizing the presentation of your results. I think it would be more compelling to first present the experimental work, then the landscape work, and then the modeling work. I would also suggest grouping them into these three

sub-headings, with sub-sub-headings if needed. That would better emphasize what you actually did and how each of the pieces built upon one another. I see why you grouped them according to general question, but I read it straight through for the first reading (to get a reader's perspective on it), and I found it very confusing to hop between approaches in the same section before I was familiar with your various approaches and methodology. Also, I think your experimental work greatly informs all other parts of your research, and as the reader, it's hard to follow the logic of some parts of the landscape analysis component without first knowing about the experimental findings.

Second, there are a few parts of your landscape analysis that seem circular, and these may require either additional analysis or additional explanation. In Li 111, you state that observations of more unvegetated areas in high hog activity marshes both supports your hog activity classification and suggests hogs generate or maintain large unvegetated areas of marsh. This seems like very circular reasoning. I thought some more detail on the methodology by which you determine the level of hog activity at each site may help clarify this, but it added to my concern. In Li 311, you said hog activity level was determined by a combination of on the ground surveys and spatial land cover information that yielded "total area disturbed m²." Then you asked if there was a relationship between hog activity and spatial land cover information, including mud cover area. It does not seem surprising there would be a positive relationship here.

Third, it isn't clear to what degree you have investigated other potential drivers of the patterns you see at the landscape scale. For example, you say found a negative correlation between hogs and mussel abundances, and hog-accessed marshes had an order of magnitude lower mussel abundance than no-hog marshes (Li 115-123). Couldn't that be evidence that hogs don't preferentially feed in marsh areas with mussels? Just to play devil's advocate, what if there's an environmental factor that makes marshes less suitable for mussels and more suitable for hogs. For example, say hogs use marshes that are less muddy, and they opportunistically eat mussels while they're there, while the mussels prefer marshes that are more muddy. It's not clear to me how you would differentiate between an environmental variable like that and hogs as a driving influence. You also found a spatial correlation in sites with and without hogs in your Sapelo Island mussel surveys (Supp Fig 5), and you provide some reasoning about why hog presence would vary spatially this way, but is also raised the question again about other potential variables (salinity, inundation, % organic matter in the mud, etc.), which may be being missed. Of course, the experimental work is a really great way to address this, but in order for your landscape survey work to stand on its own, it would be nice to see some discussion if not analysis of the degree of similarity among your various surveyed sites and the potential for any confounding factors.

Fourth, there really isn't sufficient information presented on your model to properly understand it, even in the SI. Currently, it appears only the portion related to mussel cover is included, but how this is linked to marsh recovery is only included as a reference to another paper that it says the marsh recovery model is "roughly based on." As such, it's difficult to evaluate this model and its results. Also, it wasn't clear in the main ms text how you arrived at the various forms of hog foraging behavior you used in the model. What determines where on the spectrum of foraging behavior hogs are? Does it vary over space and time? Do you have any data on this, or is this just speculative?

Finally, there were a number of smaller comments I had throughout the manuscript, which I have listed by line number below.

- Title: The term "mega-", e.g., megaherbivore, sensu Owen-Smith, is used to refer to animals that weigh >1000 kg (or 1 megagram), so I wouldn't use this prefix to describe wild hogs
- Li 40-43: The framing seems like it's written very much from an animal perspective in regards to invasive species, as I would argue many studies of plants as invasive species do actually focus on non-trophic interactions. I think there are a number of other classic case studies, e.g., beavers in S. America, that also focus on non-trophic ecosystem effects (interestingly, they mention beavers in the second paragraph but don't link to their ideas on invasion). So I think they may want to revisit some of this wording/framing
- Li 55: Should be plural: "ecosystems' ability"
- Li 69: Since the importance of this being a foundation species is a thread throughout the ms, I would provide a short explanatory phrase here of what you mean by that and what about this

species qualifies it as such.

- Li 113: I would take out the word "initial" here, as that implies preliminary data, whereas you actually have two years of surveys across 5 sites (note that the SI Fig. caption for Supp. Table 1 says 4 marshes and lists 5 sites—is this correct?)
- Li 125-131: Are half as many mussels associated with cordgrass in hog sites because there are fewer mussels, or because they grow differently at these sites? You say in Li 127 that this finding "indicates" that most mussels in marshes with hog foraging were solitary or in small aggregation—did you actually measure this?
- Li 135: hinder their ability to revegetate from what? From hog disturbance or from another disturbance?
- Li 145-149: This sentence is too long and very confusing to read, and it's a really important one. I would recommend breaking it up into 2-3 sentences. Also, the first comma should be after "experiment" rather than "exclusion", I believe.
- Li 146: This is the first time you mention crabs, which is surprising given their apparently strong links to this facilitative relationship between mussels and cordgrass and their important role in marsh ecosystems. I would at least mention them in the introduction as one of the players in the system.
- Li 151: Hogs reduce plant biomass by 48% relative to what?
- Li 194: Southeastern shouldn't be capitalized.
- Li 213: Don't need "and" after the comma
- Li 220-221: Should the word "both" come before the word "survival"? As written, it says hogs lower survival of community structure. I think you mean they lower community structure, which appears to reference the decline in crabs. However, I'm actually not actually clear what you mean by this. Do you mean the decline in diversity due to the loss of crabs, or the decline in physical structure in the community due to the loss of crab burrows?
- Li 255: should be singular "fox"
- Li 273-276: This sentence is confusing. First, it suggests there is some data out there suggesting hogs are not the cause of more fragmented marshes, but they are just more attracted to them. If so, this should be discussed explicitly in the part of the ms dealing with their interpretation of drone footage. Second, it could be written in a more straightforward way, perhaps by being broken up into 2 sentences.
- Li 305: n=30: There were 30 sites? Or 30 surveys done among 10 sites, with 3 surveys at each site?
- SI, Li 22: If there were 3 mounds per treatment per trial, and the experiment was repeated 6 times, wouldn't n=18 per treatment? And in Li 23, "generalized linear model location"?
- Fig. 1: Did you mean for the white area to overlap the drone image? It makes it very difficult to interpret. Can you just overlay the black lines on top of the image?
- Fig. 2: It looks like part of your Fig. 2 is missing.
- Fig. 4: The images of the relevant trophic levels is a nice addition to this figure, but it seems they could be better nested close to the figure, near the legend, to avoid so much empty space. Also, I would recommend using more similar and higher quality black and white art across the three sub-figures.

Response to Reviewers:

Reviewer #1 (Remarks to the Author):

The manuscript by Hensel et al. provides a very interesting example of the ways in which non-native invaders can disrupt community processes by degrading or eliminating positive interactions among key species that contribute to resilience. The study brings into line manipulative experimental studies with observations on larger spatial and temporal scales and a predictive model of recovery rates. The issue is an important one since the ability of systems like this, which are physiologically stressful to start with, to recover following anthropogenic disturbances like invasions has lessons for many other systems. In short, the authors make a strong case for introduced hogs dramatically reducing cover of cordgrass and densities of mussels. They also show that this results in the inability of either cordgrass or mussels to facilitate the other after hog related reductions.

I think the case for hogs disabling positive interactions is a good bit stronger than the case for hogs reversing the direction of the interactions or in the authors words 'flipping' the interaction. Here is where I think the authors overstate their case a bit, and I don't feel they need to do this to have an important story. I really tried to look at Fig 4A to see a reversal in the effects of mussels on cordgrass. What seems clear to me is that cordgrass biomass is substantially lower in 'Hog Control' areas. However, cordgrass abundance looks about the same for 'Mussel Addition' vs. 'Mussel Control' in those areas. Yes, ever so slightly lower with mussels, but not significantly so, and the authors should avoid making statements like "...a final plant biomass lower than unmanipulated plots" when the statistical result is NS ($p = 0.37$). Thus, I see no evidence of hogs 'flipping' the interaction. The authors also make the point in lines 156-158 that mussels in cordgrass-free areas survive longer in the presence of hogs than in cordgrass patches (and Supp Fig 3) because of the inability or unwillingness of hogs to forage in unstable sediments, it's still based on situations where both mussels and cordgrass are at low abundance, so these are not strong negative interactions. I would recommend rather than trying to make a lot of a weak case for actual reversal of positive interactions to highlight the very clear case for the complete loss of positive interactions. The compounding effect of hog predation is really to decouple this important interaction which clearly results in delayed is not complete loss of recovery. My overall conclusion is that this is a carefully conducted study with a really interesting and important message for both fundamental science and the management of salt marsh systems.

We appreciate the reviewer's comment here, which is echoed by other reviewers and the editor, and the positive and constructive comments from this reviewer as a whole. Our manuscript is clearer based on these suggestions.

We now describe the effects of hogs as a "loss" or "disabling" of positive interactions instead of a reversal or flip. We have indeed edited the title and wording in the abstract, as well as the conclusions surrounding the "flip". Even more so, after edits on Figure 4A (now Fig 2a, effects of hog exclusion/mussel experiment on cordgrass biomass) that made the point size bigger and

the figure more readable, one can easily see what the reviewer points out here: there is a clear case that hog activity causes a complete loss of positive interactions. The positive effect of mussels on biomass and crab densities inside of our cages and in other published work, but that positive effect completely disappears in caged plots that basically look the same with and without mussels. We have edited our wording accordingly.

A few additional minor areas that should be amended. One is the authors should be a bit more cautious about the conditions under which hog predation will lead to the complete collapse of the mussel population. On lines 265-267, they state “This will lead to collapse of the mussel population, severely reducing the intrinsic recovery potential of the salt marsh following drought events, which will entirely depend on recolonization from outside of the die-off area.” But recolonization from outside is an important process that could substantially delay or prevent complete die-off. Unfortunately, their very simplistic model of mussel population dynamics reflects reproduction as a simple function of current population size without any description of external recruitment. Of course mussel larvae can travel distances much greater than the spatial scale of sites with and without hogs on Sapelo Island, which based on Supplemental Fig 5 look to be separated by 10 km or less. Admittedly there is some feedback between adult population size and subsequent recruitment, since local fecundity will matter and ribbed mussels do preferentially settle in conspecific aggregations. But the prospect for mussel recovery may not be as dire unless the spatial scale of hog disturbance is a great deal larger.

We have addressed this directly in that part of the discussion (line 298), and tonally in other parts of the manuscript. We have added explanation as for how recolonization could indeed increase recovery times eventually, especially if hog densities reduce locally. “This foraging behavior will lead to a potentially permanent collapse of the mussel population, severely reducing the intrinsic recovery potential of the salt marsh following drought events, which will entirely depend on recolonization from outside of the die-off area. Mussel-driven recovery would then require a pause in hog activity and a corresponding recolonization of bare marsh, which is difficult because mussels strongly prefer to settle in conspecific aggregations. Mussels can recruit to areas that are more difficult for hogs to access (i.e., creekbanks, soft sediment) so the permanence of recovery and resilience loss depends on both recolonization and hog activity across the marsh landscape”

While we do not have data on this specific behavior, hog behavior in other systems would suggest that they move to different areas after resources are depleted locally. Thus, if hogs left and then mussels recolonized, it is possible for marsh recovery to eventually increase. This would take a long time, because mussel recruitment operates under positive density dependence as they recruit most heavily to areas of the marsh where mussels are already established (as mentioned by the reviewer). Of course, hogs could likely return to these areas once mussels came back, especially in coastal or island hog populations where home ranges may be smaller than their fully terrestrial counterparts. Regardless, mussel recolonization is a key aspect of marsh functioning and recovery on landscape scales, as recent research has shown nicely (Crotty and Angelini 2020). The sentiment from this reviewer’s final sentence here has been worked into our manuscript (line 301), as it very nicely describes the possible future of marshes and mussel-driven recovery under hog regimes.

A second issue is that I was surprised the authors never mentioned another very damaging introduced megaconsumer in the marshes of the southeastern US, the oversized rodent Nutria (*Myocastor coypus*), which has also dramatically reduced cordgrass abundance. Some mention and comparison with that invasion and the rates of recovery would be helpful, since presumably Nutria consume only cordgrass and not mussels. Thus, it would be predicted that the marshes would recover faster. Nutria has invaded large areas over decades (although controlled in many places now) so certainly there must be some useful recovery rate data to compare with this study.

We have included some comparisons with Myocastor coypus in this revision. Examples of large consumers invading marshes is rare and, in fact as the reviewer points out, the only other relevant documented example of invasive consumers in marshes is the research on nutria effects in Louisiana marshes. The idea of a comparison in recovery rates is a high-quality one, especially as it would seem that we could directly compare the differences between an invader eating grass vs grass/mussels. We have added some text on this, however, we are unable to find direct quantification of post-disturbance recovery rate inside of nutria exclusion areas. There is some evidence that nutria eradication encourages rapid recovery (Shaffer et al 2015) but these authors did not provide recovery data and recovery was confounded by increases in nutrients in the area. Another study measured nutria herbivory effects on the recovery of an LA marsh that had been overwashed by a hurricane and found that nutria did not affect the plant community (Courtemanche et al 1999). Regardless, the reviewer is correct that it would be predicted that nutria-grazed marshes would recover faster, and we have added this important context for salt marsh conservation in line 279.

“Because hog activity both directly and indirectly affects marsh recovery, regrowth of plants in hog-marshes occurs much slower than in Louisiana marshes denuded by invasive nutria (*Myocastor coypus*). Nutria can consume hectares of vegetation but, because they do not consume ribbed mussels, nutria-used marshes may be able to recover within a just year post-eradication.”

Minor fixes

Please change the Y axis in Fig 2B to reflect a real density rather than ‘per marsh’. If I read the Methods correctly, mussels were counted on 120 m transects, so possibly these marsh counts are per 120 sq m

The reviewer is correct here, and this Y axis was mislabeled. It should indeed be mussels per 120 m2 transect, and has been fixed.

Reverse the order of the entries in the legend of Fig. 3 so the order matches the lines (put Exclusion above Control). Also there is no need for colored lines (there are only two).

We have edited this figure (Now Fig 1) by making the font bigger and the lines clearer, as well as this legend edit.

Lines 384-388 describe adding 20 mussels to each of four mounds within the 2 x 2 m areas, which would be 20 mussels per sq m. (approx.). So how were mussel densities in Fig. 4B as high as 40 per sq m in the mussel addition treatments without hogs?

Great catch here. We mistakenly reported densities of mussels and crabs both per 2m², which was the size of the quadrats we used. We have edited accordingly here and in the results (line 186, 94).

Fig 4 needs to have letters A, B, C added.

We have added these letters, as well as edited this figure for general clarity with larger fonts and point sizes.

Reviewer #2 (Remarks to the Author): Summary:

The authors combine seven different studies (an observational study of landscape structure at 10 sites, two observational and three experimental studies at one location [Sapelo Island], and a population model) to explore the effects of the presence of invasive hogs on the relative abundance of ecologically important cord grass and mussel species in marshlands within the southeastern United States. Based on these varying lines of evidence, they conclude that the effect of hog predation on mussels and trampling on cordgrass is likely responsible for driving continued large-scale patterns of habitat degradation in marsh systems in the region.

The authors have generated a commendable amount of information on this topic among the various studies and data sets. In particular, the hog removal studies at Sapelo Island clearly show that hogs have a substantial effect on the recovery of degraded cordgrass habitat and that predation on mussels and likely trampling together clearly significantly reduces the presence and abundance of both species. However, I found that given the huge amount and variety of information sources, many key details of each study were missing from the manuscript (and supplement), and the various pieces were not weaved together in a convincing way to connect these local effects with large scale patterns of habitat change and resilience.

We appreciate this reviewer's comments on the content of our manuscript and for the constructive feedback on how we had woven the story together. Based on the reviewer's suggestions, we have greatly improved upon the clarity of the story, as well as added much clarification regarding our multiple approaches. We hope these edits improve the cohesion throughout the whole manuscript.

In particular, many key elements that are essential for convincing the reader that the estimation of hog 'activity' at landscape scales is robust were missing from the description of the landscape correlation analysis, and any other likely drivers of the history of fragmentation observed are not addressed (a limitation of only having 10 sampling units for statistical analysis), such as proximity to human settlements and thus other sources of disturbance.

We have added expanded site descriptions in the drone survey (line 385) and mussel survey section (line 415) of the methods, as well as a site map for the drone surveys in the Supplement. Briefly, we have now described that we specifically selected sites to maximize variation in hog activity but minimize variation in other likely drivers of marsh fragmentation by surveying marshes in the same general region of the southeastern US, of similar total size, with similar area of each cordgrass zone (i.e., tall form cordgrass on creekbanks and short form cordgrass on the marsh platform) and with similar number/size of creeks. On a landscape scale, each of these characteristics are visible when searching for sites on Google Earth and also are all indicative of marshes with similar topography, elevation, and inundation regimes. The reviewer is correct that we are unable to analytically address other possible drivers of fragmentation seen in other marshes, like runaway consumer grazing or wrack deposition, due to sample size. We point out here, and in the paper, that, regardless of the disturbance that caused the fragmentation, hog activity slows recovery from said disturbance. Marsh foundation species can form fragmented landscapes for many reasons but our experimental data and survey data taken together show that hog activity slows recovery and generates a patchy landscape that exists for much longer than when hogs are absent. Analytically, we also have accounted for possible unknown site effects in our linear mixed effects models, as we use site as our random effect in these, and all of our landscape level models. A combination of our selection of as similar sites as possible and our statistical framework, as well as how the landscape and experimental data support each other, makes us confident hog activity is responsible for site by site differences.

In addition, while linking both predation and trampling by hogs ecosystem recovery is an exciting idea with modelling potential, it does not appear that trampling is explicitly included in the model (only predation, and via three scenarios that aren't well linked to empirical data). It is not clear how mussel cover is linked to system recovery time in the paper- what is the metric of recovery used? Are the authors equating mussel increases with ecosystem recovery? If so, what evidence is there to suggest that this species alone indicates all the core functions of a recovered system?

First, we have explained the model and the predation scenarios more thoroughly in this draft in both the main text and in the supplement. Second, we more explicitly describe the predation scenarios, including a reference to the empirical data in Supp Fig 3b that we used to estimate the more unique Complete Hog Focusing predation scenario, in the form of mussel depletion rates from the Mussel Transplant Experiment in the Supplement. Mussel depletion rate showed that hogs do not slow their rate of predation on mussels when mussel mounds are partially consumed. This pattern (hogs focus on mussels, in our model) is counter to more common patterns of prey depletion, where predation pressure on remaining prey lessens at lower prey densities (incomplete hog focusing, in our model). These patterns are then used to estimate and describe the three possible predation behaviors in hogs. Additionally, these models are principally theoretical, and not based on quantifications of growth, mortality, and spatial spread. Hence the model predictions should be used in a comparative way (i.e., this is why there are no units). We have updated text to be more explicit about this in the Supplement.

As for how mussel cover is linked to system recovery time in our model, our model does this by "presuming that the cover of vegetation is proportional to that of mussels (Supplement line 32)",

which is supported by published literature. The metric of recovery is the time until vegetation recolonizes 95% of the landscape, i.e. cordgrass fills in all patches. We use this projected change in mussel population in combination with a growing body of literature shows that ribbed mussels increase marsh resilience and recovery on large-scales. Cordgrass associated with mussels is often the only patches of grass that survives drought events (~90%, Angelini et al 2016) and increasing mussel densities increases cordgrass recovery rates and multifunctionality (Angelini et al 2015, Crotty et al 2018). These citations are in our introduction when we talk about the importance of these marsh positive interactions on large scales, and we are more explicit about what “recovery” means in this revision:

“When drought-driven die-off of southern US marshes occurs in an area (from 1 to 100s of km²), cordgrass living on mussel mounds has a 98% chance at survival, while those not living on mussel mounds have a 0.01% chance⁴⁰. Once disturbance subsides, these remnant cordgrass-mussel patches become nuclei for marsh recovery during non-drought or wet years^{40,49}. With mussels present, large die-off areas recover in 2-10 years as opposed to 80 years when mussels are not present⁴⁰.” (line 77)

, and we have expanded this discussion around the model to drive home the importance of these interactions for marsh recovery and resilience (line 290). Indeed, as the reviewer suggests, mussel increases are equated with ecosystem recovery and stability in marshes, as whole marsh landscapes are more biodiverse and function at higher rates with higher mussel densities.

I also find the figures to be missing key elements that are integral to threading these disparate approaches together, particularly regarding the large-scale correlation (location and selection of the sites, regional coverage and proximity to other drivers such as human settlements), and modeling results (what are the units? How do these figures relate back explicitly to the model structure described in the supplement?).

Based on these comments and other comments from the reviewers, we have improved all of the figures. As for the large-scale correlations (i.e, drone survey and mussel survey) we have added maps and descriptions as requested by multiple reviewers. We respond to the specifics mentioned above, in the comments below.

Beyond these links across scales, it seems that the main claim of the manuscript (that positive interactions turn into antagonistic ones as a result of invasion) hangs on the result show in Figure 3A comparing cordgrass density between mussel addition and non-addition treatments in plots where hogs were present— reported as an ‘antagonistic’ interaction between mussels and cordgrass. The effect size here is very small and not significant in terms of the influence of mussels (Supp Table 5A), and occurring in the context of extremely low abundances of mussels (i.e. no difference between 3/4 treatments in Fig 3B, save the addition + exclusion treatment, which also suggests that mussels are likely not playing a functional role in these treatments). Given the small effect size and low mussel density during this effect, how likely is this to be truly an antagonist interaction (and what would the mechanisms be) vs another effect associated with changing mussel densities over the course of the experiment (e.g. extra hog trampling in the mussel addition plots), which seems far more likely?

We have edited our wording accordingly in the title, abstract, and in places throughout the ms where we use 'flip' or 'antagonistic'. We chose to use words like 'loss of positive interactions' and 'disable positive interactions' as suggested by other reviewers. Also, an edit of the figure from the hog x mussel exclusion experiment (Fig 2) making the points more readable supports what the reviewer asserts here. We did not demonstrate an antagonistic interaction and, while it is clear that both mussels and cordgrass are liabilities for each other when hogs are foraging in the marsh, our case for the disabling of positive interactions is much stronger. These comments from the reviewer have strengthened the inference from our work and we are appreciative.

Should the authors wish to retain the integration of these seven hefty data sets—which could be published convincingly in several parts with sufficient details—I believe the piece is a better fit for a longer format publication where there is space to provide the necessary details and links between the myriad studies. Below are additional detailed comments that I hope the authors will find helpful in revising their piece.

Core feedback and suggestions:

Landscape structure correlation study: Where are the sites located? How many sites have been binned into each activity group? How do the sites (and data) relate to the cited Sharp and Angelini paper cited for the method? Are these the same data presented in this manuscript, or was the same method applied to new sites? If it is the latter, far more information needs to be provided beyond the brief description in the methods (with no additional information in the supplement). I could think of other disturbance factors that could also drive patch number and size in these areas (proximity to human populations, history of drought and habitat modification). Yet no covariates are presented in the analysis; however, with only 10 sampling units it is unlikely to be possible to examine additional drivers due to statistical limitations.

We have added this information to the description of site selection for the drone survey (line 385) within the Large-Scale Geographical Survey section of the methods. To directly answer the questions posed by the reviewer here: the sites were located along the Atlantic coastline in Florida and Georgia. We did indeed select sites used in the cited Sharp and Angelini paper, but we present different data here and only use their quantification of hog activity levels as the independent variable in our analyses and figures. Including discussion of mud area as an indication of hog activity was circular and confusing. Originally, we presented this figure/results as a confirmation that the Sharp and Angelini method identified marshes that had different levels of hog activity, because "mud area per marsh" from the drone surveys was different data than their hog activity levels (i.e., estimated hog disturbance per 10000m² area). We have removed the wording that suggests that hog activity causes differences in mud area, but we left this supplementary figure in this submission and are much clearer about our site selection being from sites known to vary in hog activity from the Sharp and Angelini paper, in this draft. Other disturbance factors were not mentioned in the first draft, but we have added some text on each of the suggestions by this reviewer ("We were not able to control for distance to human populations in this survey, but no sites were within 1km of heavy human development. We selected marshes with full cordgrass monocultures (i.e., short form cordgrass in marsh platforms, tall form cordgrass along creekbanks) and similar creek size in order to minimize variation in elevation,

salinity, inundation time, mussel recruitment, and percent organic matter.” *line 391*). *Drought and habitat modification effects would not be expected to vary across the scales of this survey, but we did not control for distance to human populations and have now mentioned that as well. Additionally, we fit generalized linear mixed models with site as a random effect to incorporate potential site-level differences not controlled for in our analyses. We also have added a site map with the location of the drone sites in the Supplement.*

Hog fecal analysis: The fecal analysis survey results play a major role in the Results section of the manuscript text as evidence for the consumption of mussels in the region. However, the methods for this data collection (i.e. approach to sampling, number of samples, analytical techniques) are not included in the study Methods section, and instead given in part in the supplement only. It is also not clear how these data are used to inform the predation component of the modeling portion of the paper. The authors create three predation ‘scenarios’ (haphazard predation, incomplete focus on mussels, and complete focus on mussels), but it is not clear what supports these scenarios and how they are linked back to evidence of predation relative to other prey (presumably from the scat analysis).

We address this comment thoroughly below, under the suggestions for Supp Table 1 and fecal sample description. Most importantly, the fecal survey was not what we used to inform the modeling portion of the paper, rather to, as the reviewer states, provide evidence for the consumption of mussels in the region. This was important because, besides one observation of “mussels” (no species or family) being found in hog stomachs from a brackish marsh in the 1970s (Wood and Roark 1980), our simple feces survey is the first major piece of evidence that hogs are frequently consuming ribbed mussels in salt marshes. The context of this fecal survey is much clearer now, as we have also added text in line 194-7 to show that we used mussel consumption rates from our mussel transplant experiment as our estimates for hog predation rates. “If mussels are a primary food source for hogs- as we have shown in our empirical work, then specific predation rates increase dramatically as hogs focus heavily on the remaining mussels when they search the marsh (Fig 5a), a pattern we observed in our mussel transplant experiment (Supp Fig 3b)”.

As explained below, the feces study was actually the initial inspiration for this project and we simply recorded all of the feces we found in the marsh with mussel shells in it. We did not expand this to analyzing fecal content, nor did we quantify relative abundance, which is also why we try not to overemphasize the importance of this aspect of our work. We appreciate this comment because we are now clearer about the importance of this survey, and its context within our inference.

Mussel population dynamics: The idea of linking both predation and trampling to ecosystem recovery is an excellent one and question that could be well addressed through modeling. However, while the study states that the model links both the effects of hog trampling and predation on marsh recovery dynamics, it is not clear how trampling is incorporated into the model (no equations are provided, or translation between original model and the one used), and it is unclear how changes in mussel population size are translated into ‘marsh recovery’ specifically, which is only mentioned a few words at the end of the Methods (lines 410-411) and

not in the supplement. The graph of results (Figure 5) contains no units, and it's unclear what hog presence means.

We have addressed these comments above. We have also added lines in the Supplement regarding marsh recovery “We model how hog density H affects the cover of mussels M in a salt marsh which indirectly affects the ability of a marsh to recover from drought-induced die off. Hence, for simplicity, we presume that the cover of the vegetation is proportional to that of mussels. (line 32)”

Hog exclusion experiments: The hog exclusion experiment, and two-factor hog exclusion and mussel addition experiment are solid evidence for the effects of hogs via predation and trampling (though the effects of the two mechanisms can't be disentangled in the latter experiment). To my mind, these are core elements of the study that should be given more focused attention on their own and could easily comprise a stand alone ecologically focused manuscript.

We now present this experiment first in the results, methods, and discussion. See: Experimental Hog Exclusions section at the beginning of results and methods now.

It seems that the main thrust of the manuscript hangs on the result show in Figure 3A (thought here are no panel labels provided) comparing cordgrass density between mussel addition and non-addition treatments in plots where hogs were present— reported as an ‘antagonistic’ interaction between mussels and cordgrass. The effect size here is incredibly small, and occurring in the context of extremely low abundances of mussels (as seen in Figure 3B- no difference between 3/4 treatments, save the addition + exclusion treatment, which suggests that mussels are not playing a functional role in these treatments). Given the small effect size, and low mussel density of this effect, how likely is this to be truly an antagonist interaction (and what would the mechanisms be) vs another effect associated with changing mussel densities over the course of the experiment (e.g. extra hog trampling in the mussel addition plots)?

We have changed the wording away from a “flip” in positive interactions to simply highlight that those positive interactions may not be antagonistic, but they are definitely lost in the presence of hogs.

Additional detailed feedback:

Suggest reviewing plots to standardize text font and size/style between figures as it's quite variable.

Yes, this has been fixed now.

What are the measures of error in Figures 2 and 4?

Fig 2 and 4 (now Fig 4 and Fig 2 respectively) were graphed with standard error bars. We have added that to the caption.

Figure 1: It would be more effective to show the study region and all 10 sites and also highlighting the location of the local experiments at Sapelo Island in relation to the observational data collection. You could use two insets to demonstrate the extremes of the range.

We have changed this figure (now Fig 3) to stitched images of whole marshes that vary in hog activity, and fragmentation, instead of the small subset images previously submitted. These images are on a much larger scale and better depict the large-scale differences in how these marshes look from the drones. We have added the suggested edits above, to the site maps in the supplemental materials.

Figures 2 and 4 need panel labels.

Added.

Figure 5: What does ‘Hog presence’ mean? What are the units and scales for all axes?

We have added text to the Supplement to address these questions. Line 61-3 “These models are principally theoretical, and not based on quantifications of growth, mortality, and spatial spread. Hence the model predictions should be used in a comparative way. Parameters used in the current model were $r = 1$, $K = 100$, $d = 10$, $a = 20$ (note that a is only used in model three, partial focusing)”. Hog presence is hog density, H in our model.

Supplemental model description: How is trampling considered in the model? How is the ‘recovery time’ response variable measured (i.e. Figure 5)?

Trampling is considered in the “no hog focusing” behavior model (i.e., consistent, low predation pressure in Fig 5a, linear decreasing mussel abundance with increasing hog abundance in Fig 5b). We have made this more explicit in the methods and results sections, as well as in the Supplement. Recovery time is “how long the system would require to return to 95% of full cover if hog predation would cease (i.e., recovery time),” (Supp Materials line 57).

Supplemental Table 1 and fecal sampling description: It’s clear that hogs eat mussels, but what seems more important for linking to the modelling components is the composition of feces in each sample, rather than the ubiquity of any shell across fecal samples. How does the relative abundance (count, weight, volume) of shell compare to other diet items?

Our goal with this hog feces survey was to simply record if, and how many, of the hog feces we found in the marsh had mussel shells in them. We did not collect feces and bring back to the lab for analyses, instead opting for a quick field analysis to determine if hogs were indeed consuming mussels. This survey was actually the initial inspiration for this project, as we stumbled across mussel-filled feces while collecting data for another project in that area of Sapelo Island. This was, to our knowledge, the first observations of ribbed mussels being consumed by feral hogs and we began recording the number of feces found with and without mussels that day, and did not move into a more complex analysis. Addressing this reviewer’s

comment, we did not do a good enough job of explaining how we determined the hog predation behavior in the model. As a part of our mussel transplant experiment measuring mussel depletion rate by hog predation in cordgrass patches and adjacent bare mud, we were able to calculate predation rates that we used as estimates in that model. That rate of mussel depletion in our mussel transplant experiment is shown in Supp Fig 3b, and the reasons for selecting the rates/behavior that we did are at the end of Results, line 190-205.

Reviewer #3 (Remarks to the Author):

This is an interesting manuscript that brings together multiple approaches across a range of scales to investigate the influence of invasive hogs on marsh recovery via their disruption of a positive facilitation. The use experiments, modeling, and landscape scale surveys to show that, when hogs are absent, mussels and cordgrass have a facilitative relationship that increases the likelihood and rate of marsh recovery after a disturbance (e.g., drought).

However, in the presence of invasive hogs, this facilitation is reversed, and mussels attract hogs to forage in cordgrass clusters, which both reduces mussels and cordgrass, as well as the likelihood and rate of marsh recovery. Thus, the presence of hogs on the landscape translates into more bare mud ground and higher number of smaller patches of marsh grass. These findings have important implications for the conceptual understanding of facilitative relationships and potential impacts on invasive species, which has not been very well-studied in the literature. They also inform conservation of marsh ecosystems in the southeast by illustrating the imperative nature of hog removal for coastal resilience, which is an issue of increasing concern. Thus, I think this paper makes an important and novel contribution to the literature, and I expect this paper will be of high significance for several fields and will be well-cited. That said, there are several areas I think the manuscript could be strengthened that I think should be addressed prior to publication.

We thank the reviewer for these comments on the scope and importance of our work. Our responses below greatly strengthen the manuscript narrative and clarity.

First, the multiple approaches and suites of data presented are a great strength of the paper, but I found them to be presented in a very confusing way. Particularly in a paper with this format, where methods come after results and may never be read, I would include a brief summary statement of what you did at the start of each results section. For example, Li 145: “We conducted a factorial experiment in which we added mussels and excluded hogs to examine their effects cordgrass and marsh consumer density.” I would also suggest re-organizing the presentation of your results. I think it would be more compelling to first present the experimental work, then the landscape work, and then the modeling work. I would also suggest grouping them into these three sub-headings, with sub-sub-headings if needed. That would better emphasize what you actually did and how each of the pieces built upon one another. I see why you grouped them according to general question, but I read it straight through for the first reading (to get a reader’s perspective on it), and I found it very confusing to hop between approaches in the same section before I was familiar with your various approaches and methodology. Also, I think your experimental work greatly informs all other parts of your research, and as the reader, it’s hard to

follow the logic of some parts of the landscape analysis component without first knowing about the experimental findings.

We have implemented this suggested reorganization for clarity. We have presented the work in the order suggested by this reviewer, as well as added some brief and direct methods sentences to the results section in order to help the reader keep each part of our approach straight. Our subsections of this work are as follows:

Experimental Hog Exclusions

Hog effects on marsh recovery: 2-year patch recovery experiment

Hog effects on salt marsh resilience and positive interactions: hog x mussel experiment

Large-scale Geographical Surveys of Hog Effects across Marsh Landscapes

Hog effects on salt marsh fragmentation and recovery: drone survey

Hog effects on salt marsh resilience and positive interactions: mussel abundance survey

Modelling Marsh Recovery with Hogs

Hog effects on long-term salt marsh resilience and recovery: marsh recovery model

Each section of the results and methods now fits within these subheadings, and figure captions also reference these subsections.

Second, there are a few parts of your landscape analysis that seem circular, and these may require either additional analysis or additional explanation. In Li 111, you state that observations of more unvegetated areas in high hog activity marshes both supports your hog activity classification and suggests hogs generate or maintain large unvegetated areas of marsh. This seems like very circular reasoning. I thought some more detail on the methodology by which you determine the level of hog activity at each site may help clarify this, but it added to my concern. In Li 311, you said hog activity level was determined by a combination of on the ground surveys and spatial land cover information that yielded “total area disturbed m-2.” Then you asked if there was a relationship between hog activity and spatial land cover information, including mud cover area. It does not seem surprising there would be a positive relationship here.

We have rectified this confusing section in this revision. First, we did not make clear that the sites chosen in our drone survey were sites known to vary in hog activity because we selected several sites from the cited Sharp and Angelini 2019 paper. Because we use different response variables from that paper, we were attempting to use our mud cover graph in the Supplement as a sort-of ground truthing to compare the amount of muddy, unvegetated area quantified by our drone image analysis to the estimated “total area disturbed” variable used by Sharp and Angelini to describe variation in hog activity in marshes. Activity in that paper was estimated with an area-variable, which of course invited comparisons to our “mud area” variable in the previous draft. Regardless, the main takeaway from the drone surveys is the patchiness (patch number and patch size) generated by hogs and not the total area disturbed by hogs, so we

understand how the previous presentation was confusing and distracting. We do feel it is important to describe the scale at which we observed hog disturbance from these surveys (~500m² disturbed area), which we now just do in words now, and we do feel it may have been informative to see that our mud area measurements back up the hog activity levels from Sharp and Angelini. We however have removed any circular logic as mentioned by the reviewer, and are more explicit as to the definition of hog activity in these analyses.

Third, it isn't clear to what degree you have investigated other potential drivers of the patterns you see at the landscape scale. For example, you say found a negative correlation between hogs and mussel abundances, and hog-accessed marshes had an order of magnitude lower mussel abundance than no-hog marshes (Li 115-123). Couldn't that be evidence that hogs don't preferentially feed in marsh areas with mussels? Just to play devil's advocate, what if there's an environmental factor that makes marshes less suitable for mussels and more suitable for hogs. For example, say hogs use marshes that are less muddy, and they opportunistically eat mussels while they're there, while the mussels prefer marshes that are more muddy. It's not clear to me how you would differentiate between an environmental variable like that and hogs as a driving influence. You also found a spatial correlation in sites with and without hogs in your Sapelo Island mussel surveys (Supp Fig 5), and you provide some reasoning about why hog presence would vary spatially this way, but is also raised the question again about other potential variables (salinity, inundation, % organic matter in the mud, etc.), which may be being missed. Of course, the experimental work is a really great way to address this, but in order for your landscape survey work to stand on its own, it would be nice to see some discussion if not analysis of the degree of similarity among your various surveyed sites and the potential for any confounding factors.

*We have added this information to the description of site selection for the drone survey within the Large Scale Geographical Survey section of the methods. Other disturbance factors were not mentioned in the first submission, but we have added text on each of the suggestions by this reviewer and reviewer 2 (line 389). Variables mentioned here could indeed affect these processes and we did take steps to ensure that there was not large variation in these both within our drone surveys and mussel abundance surveys. Salinity, inundation time, and percent organic matter would not vary much between our sites, as indicated by the similar species composition (i.e., *Spartina alterniflora* monoculture with tall form *Spartina* along creek banks and short form *Spartina* on the marsh platform). Additionally, we surveyed marshes with similar distances to large creeks/rivers, as an attempt to reduce variation in inundation time and mussel recruitment potential. We have now mentioned all of this in the methods section (line 394). “We selected marshes with full cordgrass monocultures (i.e., short form cordgrass in marsh platforms, tall form cordgrass along creekbanks) and similar creek size in order to minimize variation in elevation, salinity, inundation time, mussel recruitment, and percent organic matter. “ Additionally, we fit generalized linear mixed models with site as a random effect to cover potential site-level differences not accounted for in our analyses. As mentioned by the reviewer, our experimental work is a great way to address these potential gaps in inference in the landscape analyses (not to mention, the experiments were conducted in similar marshes).*

Fourth, there really isn't sufficient information presented on your model to properly understand it, even in the SI. Currently, it appears only the portion related to mussel cover is included, but how this is linked to marsh recovery is only included as a reference to another paper that it says the marsh recovery model is "roughly based on." As such, it's difficult to evaluate this model and its results. Also, it wasn't clear in the main ms text how you arrived at the various forms of hog foraging behavior you used in the model. What determines where on the spectrum of foraging behavior hogs are? Does it vary over space and time? Do you have any data on this, or is this just speculative?

We have explained the model and the predation scenarios more thoroughly in this draft in both the main text and in the supplement. We more explicitly describe the predation scenarios, including a reference to the empirical data in Supp Fig 3b that we used to estimate the more unique Complete Hog Focusing predation scenario, in the form of mussel depletion rates from the Mussel Transplant Experiment in the Supplement. Mussel depletion rate showed that hogs do not slow their rate of predation on mussels when mussel mounds are partially consumed. This pattern (hogs focus on mussels, in our model) is counter to more common patterns of prey depletion, where predation pressure on remaining prey lessens at lower prey densities (incomplete hog focusing, in our model). These patterns are then used to estimate and describe the three possible predation behaviors in hogs (line 447).

Finally, there were a number of smaller comments I had throughout the manuscript, which I have listed by line number below.

- Title: The term "mega-", e.g., megaherbivore, sensu Owen-Smith, is used to refer to animals that weigh >1000 kg (or 1 megagram), so I wouldn't use this prefix to describe wild hogs

We have revised this term to "large consumer" and actually revised this term in another manuscript about hogs. We were operating with the definition from Martin and Klein 1989 that defined megaconsumers as >45kg and a recent review in ProcB (Moleón et al 2020) provided some nice evidence for animals like hogs being 'keystone megafauna' which has no direct tie to a mass limit. We do think that hogs represent this role in marshes, especially as they are one of the largest marsh organisms, but feel no need to distract from the focus of our manuscript with a potentially confusing term. Thank you for this suggestion

- Li 40-43: The framing seems like it's written very much from an animal perspective in regards to invasive species, as I would argue many studies of plants as invasive species do actually focus on non-trophic interactions. I think there are a number of other classic case studies, e.g., beavers in S. America, that also focus on non-trophic ecosystem effects (interestingly, they mention beavers in the second paragraph but don't link to their ideas on invasion). So I think they may want to revisit some of this wording/framing

Our closing sentence was too broad here, and we have changed "invasive species" to "invasive consumers" as to not invite comparisons to plant invasions. We have added a mention of invasive beavers as an example of invasive consumers changing native food webs as well, to be more complete

- Li 55: Should be plural: “ecosystems’ ability”

Corrected

- Li 69: Since the importance of this being a foundation species is a thread throughout the ms, I would provide a short explanatory phrase here of what you mean by that and what about this species qualifies it as such.

*We have added relevant text and citations to this sentence (line 71 now). “..foundation species *Spartina alterniflora* (hereafter cordgrass) that forms the monocultures that structurally define intertidal marsh habitat”*

- Li 113: I would take out the word “initial” here, as that implies preliminary data, whereas you actually have two years of surveys across 5 sites (note that the SI Fig. caption for Supp. Table 1 says 4 marshes and lists 5 sites—is this correct?)

Corrected. Actually, the hog-mussel predation reported in this manuscript was first discovered when we haphazardly found hog feces with mussel shells in it in the marsh while taking data for another experiment, spurring many of the questions addressed here. That is why we considered this data collection initial! Fixed now.

- Li 125-131: Are half as many mussels associated with cordgrass in hog sites because there are fewer mussels, or because they grow differently at these sites? You say in Li 127 that this finding “indicates” that most mussels in marshes with hog foraging were solitary or in small aggregation—did you actually measure this?

Yes, we did actually measure the number of singleton mussels (i.e., mussels not in mounds) and quantified the % of mussels on defunct mounds. We now mention those numbers directly in line 231 and have added that column to the mussel survey table ST 5. Briefly, we counted 96 total singleton mussels in hog-accessed marshes along creekheads and 27 on the marsh platform. We only counted 3 and 12 singleton mussels in those respective marsh locations in the non-hog accessed marshes.

- Li 135: hinder their ability to revegetate from what? From hog disturbance or from another disturbance?

We do actually mean both here, and have added that text. Marsh recovery from any disturbance (e.g., hog disturbance, drought, or wrack deposition), most often proceeds in the same way, with regrowth from patches powered by patches associated with ribbed mussels. Thus, hog activity hinders cordgrass revegetation after any event causes plant dieback.

- Li 145-149: This sentence is too long and very confusing to read, and it's a really important one. I would recommend breaking it up into 2-3 sentences. Also, the first comma should be after "experiment" rather than "exclusion", I believe.

*We have broken up this sentence (lines 122-130), which indeed is the most important line from our factorial experiment, into several to improve clarity. "When hogs are absent, mussels and cordgrass facilitate each other and increase bioturbating crab densities (*Uca spp.*, *Sesarma reticulatum*, *Panopeus herbstii*, *Eurytium SPP*)²⁴. Similar to findings in other experimental work^{40,51}, mussel additions increase cordgrass biomass by 1.5x (Fig 2a), mussel survival is high (Fig 2b), and crab densities are tripled (Fig 2c) compared to our no mussel plots.*

In uncaged plots, hogs completely disable the positive plant-mussel interactions: hogs reduce plant biomass by 48% in mussel addition treatments compared to the caged mussel addition plots where positive interactions are intact (estimate: -298.7 g-cordgrass/m² difference; Fig. 2a, Supp Table 3; Hog*Mussels: $F_{1,35} = 5.86$, $p = 0.03$). The uncaged mussel addition plots were frequently disturbed and trampled, creating a final plant biomass equivalent to or slightly lower than uncaged plots with no mussels (estimate: -98.9 g-cordgrass/m², $t = 2.8$ $p = 0.37$)."

- Li 146: This is the first time you mention crabs, which is surprising given their apparently strong links to this facilitative relationship between mussels and cordgrass and their important role in marsh ecosystems. I would at least mention them in the introduction as one of the players in the system.

A line pointing out that mussel-cordgrass mutualism also attracts bioturbating crabs has been added to the introduction, in the same area where the reviewer suggested we elaborate on foundation species (line 76 "In this mutualism, cordgrass facilitates mussels by providing settlement substrate and reducing temperatures via shading, while mussels in mounds enhance grass success, attract bioturbating burrowing crabs, and protect grass from drought through amelioration of edaphic soil stress")

- Li 151: Hogs reduce plant biomass by 48% relative to what?

We have reworded this line to be more clear (line 124). "Similar to findings in other experimental work^{40,51}, mussel additions increase cordgrass biomass by 1.5x (Fig 2a), mussel survival is high (Fig 2b), and crab densities are tripled (Fig 2c) compared to our no mussel plots."

- Li 194: Southeastern shouldn't be capitalized.
- Li 213: Don't need "and" after the comma

Above two errors fixed

- Li 220-221: Should the word “both” come before the word “survival”? As written, it says hogs lower survival of community structure. I think you mean they lower community structure, which appears to reference the decline in crabs. However, I’m actually not actually clear what you mean by this. Do you mean the decline in diversity due to the loss of crabs, or the decline in physical structure in the community due to the loss of crab burrows?

We have edited this sentence to read “Hog activity lowers the survival of both foundation species (Fig 4a,b) and alters community structure (Fig 4c), which reduces long-term resilience and slows large-scale recovery”

- Li 255: should be singular “fox”

Corrected

- Li 273-276: This sentence is confusing. First, it suggests there is some data out there suggesting hogs are not the cause of more fragmented marshes, but they are just more attracted to them. If so, this should be discussed explicitly in the part of the ms dealing with their interpretation of drone footage. Second, it could be written in a more straightforward way, perhaps by being broken up into 2 sentences.

This sentence has been edited for clarity. Referencing the “generate or attracted to” phrase, we were repeating a point from the cited paper that suggests hogs either generate or are attracted to lower elevation marshes. The wording made it seem like hogs may not generate fragmented marshes, which our drone surveys clearly suggest they do! This long sentence has been broken up to more clearly convey our point: when hogs use marshes, those marshes are more fragmented, lower in elevation, and have lower levels of ecosystem functioning because hogs reduce the densities of the multifunctional marsh residents. Line 306 “Thus, as hogs access more areas throughout the coastal US, valuable habitats will be more fragmented and lower in elevation”⁸². Marshes with high hog activity will also have lower levels of ecosystem functioning since marsh multifunctionality can be controlled by a few, super abundant species like ribbed mussels and burrowing crabs^{24,83} that all decline in the presence of hogs (Fig 2).”

- Li 305: n=30: There were 30 sites? Or 30 surveys done among 10 sites, with 3 surveys at each site?

This was an incorrectly explained part of the methods, and we appreciate the reviewer catching this mistake. 28 surveys were done among 14 sites, with two drone flights (i.e., surveys) conducted per site, and the means of each metric summarized per site (n = 14). However, we eventually binned these sites based on hog activity levels, so our n = 5, 6, and 3 for each of high, medium, and low hog activity levels. This is now updated to be clear and accurate.

- SI, Li 22: If there were 3 mounds per treatment per trial, and the experiment was repeated 6 times, wouldn't n=18 per treatment? And in Li 23, "generalized linear model location"?

Yes, this mistake has been corrected. n=18. Line 23 had a typo, the word "location", which is also corrected.

- Fig. 1: Did you mean for the white area to overlap the drone image? It makes it very difficult to interpret. Can you just overlay the black lines on top of the image?

Yes that must have been some sort of file conversion error. We have actually replaced this figure with larger, whole-marsh pictures from our drone surveys that nicely capture the scale of the patchiness of high vs low hog activity marshes.

- Fig. 2: It looks like part of your Fig. 2 is missing.

Not sure what happened here but we have fixed this up!

- Fig. 4: The images of the relevant trophic levels is a nice addition to this figure, but it seems they could be better nested close to the figure, near the legend, to avoid so much empty space. Also, I would recommend using more similar and higher quality black and white art across the three sub-figures.

We have made these suggested changes.

We appreciate your time and consideration of our work.

Sincerely,

Marc Hensel
Department of Biology
University of Massachusetts Boston
Boston, MA,
Marc.Hensel001@umb.edu

Peer Review, second round –

Reviewer #1 (Remarks to the Author):

The authors have done a good job responding to my comments. The manuscript has shifted its focus away from flipping positive interactions to the strength of the story. Key comparisons have been clarified and included some modest comparison with Nutria. Other minor issues have also been modified or fixed. The manuscript is generally much improved with a tighter focus on a more strongly supported story that is an important one.

Reviewer #3 (Remarks to the Author):

I think this is an important, thorough, and well-written manuscript, and I don't have any further suggestions at this stage.

I appreciated your thoughtful replies to my inquiries and suggestions, and I thought you all did a good job addressing my concerns, as well as those of the other reviewers. I really liked the reorganization of the paper, and I think it makes your data and logic much easier to follow. Sections on methods, site selection, and model development all benefitted from increased clarity and detail. I also appreciated the increased focus on dismantling of positive interactions rather than flipping them into antagonistic interactions, although I expect there might be some interesting areas there for future work.

Regarding use of the term "megaconsumer", I wasn't familiar with the Moleón et al. paper, so thank you for bringing that to my attention. Given the recommendations in that paper, I think you probably could use the term "megafauna" if you felt it was more appropriate than "large consumer", provided you cite the Moleón et al. paper and follow their recommendation to define how you're using the term and the "logic underpinning... [your] definition." However, I agree with your assessment that the term "megaconsumer" may be more likely to distract or confuse a reader. I also like the simplicity of "large consumer."